# FEEDBACK-DRIVEN RECURRENT QUANTUM NEURAL NETWORK UNIVERSALITY

**Lukas Gonon**[*]
School of Computer Science
University of St. Gallen, Switzerland
lukas.gonon@unisg.ch

**Rodrigo Martínez-Peña**
Donostia International Physics Center
San Sebastián, Spain
rodrigo.martinez@dipc.org

**Juan-Pablo Ortega**
School of Physical and Mathematical Sciences
Nanyang Technological University, Singapore
Juan-Pablo.Ortega@ntu.edu.sg

## ABSTRACT

Quantum reservoir computing uses the dynamics of quantum systems to process temporal data, making it particularly well-suited for machine learning with noisy intermediate-scale quantum devices. Recent developments have introduced feedback-based quantum reservoir systems, which process temporal information with comparatively fewer components and enable real-time computation while preserving the input history. Motivated by their promising empirical performance, in this work, we study the approximation capabilities of feedback-based quantum reservoir computing. More specifically, we are concerned with recurrent quantum neural networks, which are quantum analogues of classical recurrent neural networks. Our results show that regular state-space systems can be approximated using quantum recurrent neural networks without the curse of dimensionality and with the number of qubits only growing logarithmically in the reciprocal of the prescribed approximation accuracy. Notably, our analysis demonstrates that quantum recurrent neural networks are universal with linear readouts, making them both powerful and experimentally accessible. These results pave the way for practical and theoretically grounded quantum reservoir computing with real-time processing capabilities.

## 1 INTRODUCTION

Recent advances in quantum computing have led to a rapid development of quantum machine learning methods. These methods aim to exploit the potential computational speed-up and reduced complexity offered by quantum computing for machine learning purposes. For learning problems with temporal structure, quantum reservoir computing (QRC) has emerged as a promising approach for exploiting noisy intermediate-scale quantum (NISQ) technologies. In contrast to classical machine learning methods based on bits valued in $\{0, 1\}$, quantum bits (qubits) can be in a continuum of states. QRC aims to exploit this fundamental difference to build efficient machine learning methods for time series prediction and learning.

In this paper, we are concerned with recurrent quantum neural networks (RQNN), a particular type of quantum reservoir computing method. RQNNs are a quantum analogue to classical recurrent neural networks. RQNNs are built from quantum neural networks (QNNs), with weights and biases typically realized via quantum circuits. Thus, these networks can be evaluated directly on quantum computers. Thereby, quantum machine learning aims to achieve a significant increase in neural network expressivity and computational speed-up in inference and training.

---

[*]Also affiliated as Honorary Senior Lecturer with the Department of Mathematics, Imperial College, London, United Kingdom

Motivated by their promising empirical performance, in this work, we study the approximation capabilities of feedback-based quantum reservoir computing methods and, specifically, RQNNs. In particular, our work provides precise bounds on the number of qubits and the size of the underlying quantum circuit that is required to guarantee a prescribed approximation accuracy. Our results show that QRNNs can approximate regular state-space systems using a quantum circuit with qubit number only growing logarithmically in the reciprocal of the prescribed approximation accuracy and with error rates not suffering from the curse of dimensionality. Thereby, our results pave the way for theoretically grounded quantum reservoir computing with real-time processing capabilities.

## 1.1 Related Literature

Quantum reservoir computing methods have been extensively studied for a variety of time-series prediction and learning tasks, employing different architecture types such as online protocols (Mujal et al., 2023; Franceschetto et al., 2024), mid-circuit measurements and reset operations (Hu et al., 2024; Murauer et al., 2025), feedback protocols (Kobayashi et al., 2024), QRC with quantum memristors (Spagnolo et al., 2022) and hybrid QRC techniques (Pfeffer et al., 2022; 2023). We provide a detailed discussion of QRC methods in Appendix A.

Despite these promising developments, key questions regarding universal approximation capabilities and expressivity of feedback-driven QRC methods have not been addressed in the literature. For classical neural networks, qualitative and quantitative universal approximation theorems have been extensively studied, with seminal works including, e.g. Hornik (1991); Barron (1993); Yarotsky (2017). Universality results for the dynamic reservoir computing setting have been obtained in (Grigoryeva & Ortega, 2018a;b; Gonon & Ortega, 2020; 2021; Gonon et al., 2023) for echo state networks, state-affine systems and linear systems with polynomial / neural network readouts. For (feedforward) QNNs first qualitative results on universal approximation properties of QNNs have been proved only very recently Pérez-Salinas et al. (2020); Schuld et al. (2021). Subsequently, quantitative approximation error bounds for feedforward QNNs were proved in Gonon & Jacquier (2025); Yu et al. (2024); Aftab & Yang (2024).

For RQNNs, no quantitative approximation error bounds have been previously available in the literature. Moreover, previous universality results concerning QRC models have relied on the use of polynomial output layers (Chen & Nurdin, 2019; Chen et al., 2020; Nokkala et al., 2021; Sannia et al., 2024b;a), which yield a polynomial algebra that can then be used with the Stone-Weierstrass theorem to obtain universality statements. Nevertheless, most numerical and experimental implementations of reservoir computers use linear output layers due to their simplicity and fast training.

## 1.2 Contributions

For applications of QRC methods in learning tasks with temporal dependence, a precise understanding of RQNN approximation capabilities is essential. In this paper, we derive approximation error bounds and prove universality statements for RQNN families with a linear output layer and in the context of the feedback protocol. Universality refers to the ability of these families to uniformly approximate arbitrarily well a large category of dynamic processes, so-called fading memory input/output systems. Thereby, we contribute to a precise understanding of RQNN approximation capabilities in several aspects.

- We provide RQNN approximation error bounds for regular state-space systems. Our first main result, Theorem 4.6, shows that RQNNs are able to approximate regular state-space systems without the curse of dimensionality, using quantum circuits with qubit number only growing logarithmically in the reciprocal of the prescribed approximation accuracy.

- In our second main result, Theorem 4.8, we prove that RQNNs can uniformly approximate the arbitrary fading memory, causal, and time-invariant filters. In particular, RQNNs have approximation properties as competitive as those of popular reservoir computing/state-space system families like echo state networks, state-affine systems, or linear systems with polynomial/neural network readouts.

- To prove these results, we first derive novel qualitative and quantitative approximation error results for using feedforward QNNs to approximate functions and their derivatives (see Proposition 4.4 and Corollary 4.5).

In comparison to Gonon & Jacquier (2025), our RQNNs introduce memory through a feedback loop. Mathematically analysing our RQNNs architecture hence requires a novel, intricate analysis of QNN approximations of functions jointly with their derivatives. Moreover, approximation analysis in the temporal domain is inherently much more challenging due to the feedback loop. Proving Theorems 4.6 and 4.8 thus requires new techniques specifically tailored to deal with this situation (see Appendix C). Most previous literature on RC and QRC universality (Grigoryeva & Ortega, 2018a;b; Gonon & Ortega, 2020; 2021; Chen & Nurdin, 2019; Chen et al., 2020; Nokkala et al., 2021; Sannia et al., 2024b;a) implicitly assumes the search for an optimal model within a class in which all parameters are estimated. Also our results are formulated for variational quantum circuits for which all parameters are trainable. Nevertheless, the obtained results and developed proof techniques also promise to be useful for QRC systems in which certain parameters in the recurrent layer are randomly generated. Our RQNN architecture builds on and extends the feedforward QNN architecture introduced in Gonon & Jacquier (2025), which also admits results for the randomized setting. Hence, combining the techniques developed here with these randomized architectures may provide fruitful for studying randomization in the dynamic quantum reservoir computing setting. Moreover, the obtained approximation error bounds may serve as a crucial ingredient for bounding the overall generalization error of QRC methods, by combining our results with suitable risk bounds as obtained in other contexts in Gonon et al. (2020); Chmielewski et al. (2025).

## 1.3 OUTLINE

The paper is structured as follows. Section 2 introduces background on filters, functionals, fading-memory and echo state properties. Section 3 describes the RQNN model, a recurrent QNN with state feedback. Section 4.1 derives QNN approximation error bounds for functions and their first derivatives. We then use these results (see Proposition 4.4 and Corollary 4.5) to study the properties of the RQNN state maps for approximating more general state equations. These results are then used in Section 4.2 to prove the universal uniform approximation properties of the filters associated with RQNN systems. More specifically, in Theorem 4.6 we provide filter approximation bounds that show that RQNNs can uniformly approximate the filters induced by any contracting Barron-type state-space system. Finally, Theorem 4.8 of Section 4.3 extends this universality property to the much larger category of arbitrary fading memory, causal, and time-invariant filters. The paper concludes with Section 5, where the main contributions and outlook of the paper are summarized.

## 2 BACKGROUND ON FILTERS AND FUNCTIONALS

We start by introducing the input-output maps to be learnt in the dynamic setting. In a static context, input-output maps are given by functions of the form $f : \mathbb{R}^d \to \mathbb{R}^m$. For learning with temporal dependence, the relevant input-output maps are *filters* and *functionals* defined on sequences.

Specifically, let $(\mathbb{R}^n)^{\mathbb{Z}}$ denote the set of infinite real sequences of the form $\underline{z} = (\ldots, \boldsymbol{z}_{-1}, \boldsymbol{z}_0, \boldsymbol{z}_1, \ldots)$, $\boldsymbol{z}_i \in \mathbb{R}^n$, $i \in \mathbb{Z}$; $(\mathbb{R}^n)^{\mathbb{Z}_-}$ is the subspace consisting of left infinite sequences: $(\mathbb{R}^n)^{\mathbb{Z}_-} = \{\underline{z} = (\ldots, \boldsymbol{z}_{-2}, \boldsymbol{z}_{-1}, \boldsymbol{z}_0) \mid \boldsymbol{z}_i \in \mathbb{R}^n, i \in \mathbb{Z}_-\}$. Analogously, $(D_n)^{\mathbb{Z}}$ and $(D_n)^{\mathbb{Z}_-}$ stand for infinite and semi-infinite sequences, with elements in the subset $D_n \subset \mathbb{R}^n$. Let $D_n \subset \mathbb{R}^n$ and $B_N \subset \mathbb{R}^N$. We refer to the maps of the type $U : (D_n)^{\mathbb{Z}} \longrightarrow (B_N)^{\mathbb{Z}}$ as **filters** and to those like $H : (D_n)^{\mathbb{Z}} \longrightarrow B_N$ (or $H : (D_n)^{\mathbb{Z}_-} \longrightarrow B_N$) as **functionals**. A filter $U : (D_n)^{\mathbb{Z}} \longrightarrow (B_N)^{\mathbb{Z}}$ is called **causal** when for any two elements $\boldsymbol{z}, \boldsymbol{w} \in (D_n)^{\mathbb{Z}}$ that satisfy that $\boldsymbol{z}_\tau = \boldsymbol{w}_\tau$ for any $\tau \leq t$, for a given $t \in \mathbb{Z}$, we have that $U(\boldsymbol{z})_t = U(\boldsymbol{w})_t$. Let $T_\tau : (D_n)^{\mathbb{Z}} \longrightarrow (D_n)^{\mathbb{Z}}$, $\tau \in \mathbb{Z}$ be the **time delay** operator defined by $T_\tau(\boldsymbol{z})_t := \boldsymbol{z}_{t-\tau}$. The filter $U$ is called **time-invariant** when it commutes with the time delay operator, that is, $T_\tau \circ U = U \circ T_\tau$, for any $\tau \in \mathbb{Z}$, with the two operators $T_\tau$ defined in the appropriate sequence spaces. Finally, there is a bijection between causal time-invariant filters and functionals on $(D_n)^{\mathbb{Z}_-}$, and we can use them interchangeably (Grigoryeva & Ortega, 2018b).

A specific class of filters is given by state-space systems (such as recurrent neural networks) determined by two maps, namely the **recurrent** layer or the **state map** $F : \mathbb{R}^N \times \mathbb{R}^n \longrightarrow \mathbb{R}^N$, $n, N \in \mathbb{N}$, and a **readout** or **observation** map $h : \mathbb{R}^N \to \mathbb{R}^m$, $m \in \mathbb{N}$, given by

$$
\begin{aligned}
\boldsymbol{x}_t &= F(\boldsymbol{x}_{t-1}, \boldsymbol{z}_t), \\
\mathbf{y}_t &= h(\boldsymbol{x}_t),
\end{aligned}
\tag{1}
$$

where $t \in \mathbb{Z}$, $\boldsymbol{z}_t$ denotes the input, $\boldsymbol{x}_t \in \mathbb{R}^N$ is the state vector, and $\mathbf{y}_t \in \mathbb{R}^m$ is the output vector.

Consider now subsets $B_N \subset \mathbb{R}^N$ and $D_n \subset \mathbb{R}^n$ and a recurrent layer defined on them, that is, $F : B_N \times D_n \longrightarrow B_N$ and $h : B_N \to \mathbb{R}^m$. Denote by $D_m := h(B_N) \subset \mathbb{R}^m$. The recurrent system $F$ is said to have the ***echo state property*** with respect to inputs in $(D_n)^{\mathbb{Z}}$ when for any $\underline{z} \in (D_n)^{\mathbb{Z}}$ there exists a unique element $\underline{x} \in (B_N)^{\mathbb{Z}}$ that satisfies the first equation in (1), for each $t \in \mathbb{Z}$. When the echo state property holds, a unique filter $U^F : (D_n)^{\mathbb{Z}} \longrightarrow (B_N)^{\mathbb{Z}}$ can be associated to the recurrent system determined by $F$, namely, $U^F(\boldsymbol{z})_t := \boldsymbol{x}_t \in B_N$, for all $t \in \mathbb{Z}$. We will denote by $U_h^F : (D_n)^{\mathbb{Z}} \longrightarrow (D_m)^{\mathbb{Z}}$ the corresponding filter determined by the entire recurrent system, that is, $U_h^F(\boldsymbol{z})_t = h\left(U^F(\boldsymbol{z})_t\right) := \mathbf{y}_t \in D_m$, for all $t \in \mathbb{Z}$. The filters $U^F$ and $U_h^F$ are causal and time-invariant by construction. The echo state property is much related with the so-called ***fading memory property*** defined as the continuity of $U_h^F$ with respect to weighted norms in its domain and codomain (Boyd & Chua, 1985) or the product topologies when $D_n$ and $D_m$ are compact (Grigoryeva & Ortega, 2018b). It can be shown that when $D_m$ is compact, the echo state property implies the fading memory property (Manjunath, 2020; Ortega & Rossmannek, 2025b); see Ortega & Rossmannek (2025c) for a comprehensive account of the dynamical implications of the fading memory property as well as Ortega & Rossmannek (2025a) for a stochastic version.

## 3 RECURRENT QUANTUM NEURAL NETWORK ARCHITECTURE

Before going into details about the considered RQNN architecture, let us first explain the basic working principle of feedforward QNNs built in quantum circuits. A QNN is built by transforming quantum bits (*qubits*) in a parametric quantum circuit. Each qubit is in state $|\psi\rangle = \alpha |0\rangle + \beta |1\rangle$ for some $\alpha \in \mathbb{C}$, $\beta \in \mathbb{C}$ with $|\alpha|^2 + |\beta|^2 = 1$ and with elementary quantum bit states $|0\rangle$ and $|1\rangle$. For a circuit with $\mathfrak{n}$ qubits, at any given point in the circuit, the circuit state can thus be identified with a vector in $\mathbb{C}^{n_U}$ for $n_U = 2^{\mathfrak{n}}$. The quantum state $|\psi\rangle$ can be transformed by applying a *quantum gate*, that is, a unitary matrix $U \in \mathbb{C}^{n_U \times n_U}$. A QNN now applies quantum gates $U(\boldsymbol{x}, \boldsymbol{\theta})$ that depend on the initial data and neural network parameters and transforms the circuit accordingly. The QNN output is obtained by measuring the final quantum state after applying the circuit quantum gates.

Next, we introduce in detail the employed RQNN architecture. Our recurrent quantum circuit is constructed based on two parametric quantum gates $U$ and $V$, which we now introduce. The construction extends the feedforward QNN architecture introduced in Gonon & Jacquier (2025) to a recurrent setting by feeding back the network's state.

**Construction of $U$.** For $\delta, \gamma \in [0, 2\pi]$ and $\alpha \in \mathbb{R}$, denote by $R_x(\delta)$, $R_y(\gamma)$, and $R_z(\alpha)$ the rotations around the X-, Y-and the Z-axis, corresponding to angles $\delta$, $\gamma$ and $\alpha$, respectively, and obtained as the exponentials of the Pauli matrices:

$$R_x(\delta) := \begin{pmatrix} \cos\left(\frac{\delta}{2}\right) & -i\sin\left(\frac{\delta}{2}\right) \\ -i\sin\left(\frac{\delta}{2}\right) & \cos\left(\frac{\delta}{2}\right) \end{pmatrix}, R_y(\gamma) := \begin{pmatrix} \cos\left(\frac{\gamma}{2}\right) & -\sin\left(\frac{\gamma}{2}\right) \\ \sin\left(\frac{\gamma}{2}\right) & \cos\left(\frac{\gamma}{2}\right) \end{pmatrix}, R_z(\alpha) := \begin{pmatrix} e^{-i\frac{\alpha}{2}} & 0 \\ 0 & e^{i\frac{\alpha}{2}} \end{pmatrix}.$$

For a given accuracy parameter $n \in \mathbb{N}$, consider weights $\boldsymbol{a} = (\boldsymbol{a}^1, \dots, \boldsymbol{a}^n) \in (\mathbb{R}^{d+N})^n$, $\boldsymbol{b} = (b^1, \dots, b^n) \in \mathbb{R}^n$ and $\boldsymbol{\gamma} = (\gamma^1, \dots, \gamma^n) \in [0, 2\pi]^n$. For $i = 1, \dots, n$, we define parametric gate maps $U_1^{(i)} : \mathbb{R}^N \times \mathbb{R}^d \to \mathbb{C}^{2 \times 2}$ that map a current system state $\boldsymbol{x}$ and a current observation $\boldsymbol{z}$ to a rotation gate. Gate map $i$ depends on parameters $\boldsymbol{a}^i, b^i$ and is defined by

$$U_1^{(i)}(\boldsymbol{x}, \boldsymbol{z}) := H\, R_z\left(-b^i\right) R_z\left(-a_{N+d}^i z_d\right) \cdots R_z\left(-a_{N+1}^i z_1\right) R_z\left(-a_N^i x_N\right) \cdots R_z\left(-a_1^i x_1\right) H$$

for any $\boldsymbol{x} = (x_1, \dots, x_N) \in \mathbb{R}^N$ and $\boldsymbol{z} = (z_1, \dots, z_d) \in \mathbb{R}^d$, with $H$ the Hadamard gate. We may rewrite

$$U_1^{(i)}(\boldsymbol{x}, \boldsymbol{z}) = R_x\left(\delta^i\right), \quad \delta^i := -b^i - a_{N+d}^i z_d \cdots - a_{N+1}^i z_1 - a_N^i x_N \cdots - a_1^i x_1.$$

Moreover, we also define the gates $U_2^{(i)} := R_y\left(\gamma^i\right)$ and denote the circuit parameters by $\boldsymbol{\theta} = (\boldsymbol{a}^i, b^i, \gamma^i)_{i=1,\dots,n} \in \boldsymbol{\Theta} := (\mathbb{R}^{d+N} \times \mathbb{R} \times [0, 2\pi])^n$.

With these notations, we are now ready to define the key element of our parametric quantum circuit, the gate $U := U_{\boldsymbol{\theta}}(\boldsymbol{x}, \boldsymbol{z})$. $U$ is defined as a block matrix built from the gates $\bar{U}^{(i)}(\boldsymbol{x}, \boldsymbol{z}) = U_1^{(i)}(\boldsymbol{x}, \boldsymbol{z}) \otimes$

$U_2^{(i)}$ as follows:

$$
U_{\boldsymbol{\theta}}(\boldsymbol{x}, \boldsymbol{z}) := \begin{bmatrix} \bar{U}^{(1)}(\boldsymbol{x},\boldsymbol{z}) & \mathbf{0}_{4\times4} & \mathbf{0}_{4\times4} & \cdots & \mathbf{0}_{4\times4} & \mathbf{0}_{4\times n_0} \\ \mathbf{0}_{4\times4} & \bar{U}^{(2)}(\boldsymbol{x},\boldsymbol{z}) & \mathbf{0}_{4\times4} & \cdots & \mathbf{0}_{4\times4} & \vdots \\ \vdots & & \ddots & & \vdots & \vdots \\ \mathbf{0}_{4\times4} & \cdots & \mathbf{0}_{4\times4} & \bar{U}^{(n-1)}(\boldsymbol{x},\boldsymbol{z}) & \mathbf{0}_{4\times4} & \vdots \\ \mathbf{0}_{4\times4} & \cdots & \cdots & \mathbf{0}_{4\times4} & \bar{U}^{(n)}(\boldsymbol{x},\boldsymbol{z}) & \mathbf{0}_{4\times n_0} \\ \mathbf{0}_{n_0\times4} & \cdots & \cdots & \cdots & \mathbf{0}_{n_0\times4} & \mathbf{1}_{n_0\times n_0} \end{bmatrix}.
$$

Here, $n_0$ is chosen as the smallest natural number such that the matrix dimension $n_U = 4n + n_0$ is a power of 2, that is, $n_U = 2^{\mathfrak{n}}$. It can be easily shown that $n_0 = 4\kappa$ with $\kappa \in \mathbb{N}$, since $4n + n_0$ and $2n + n_0/2$ must be even for $\mathfrak{n} \geq 2$. Then, $U \in \mathbb{C}^{n_U \times n_U}$ is a unitary quantum gate operating on $\mathfrak{n} = \log_2(n_U) = 2 + \log_2(n + \kappa) = \lceil \log_2(2n) \rceil$ qubits with a diagonal-block structure:

$$
U_{\boldsymbol{\theta}}(\boldsymbol{x}, \boldsymbol{z}) = \sum_{i=0}^{n-1} |i\rangle \langle i| \otimes \bar{U}^{(i+1)}(\boldsymbol{x},\boldsymbol{z}) + \sum_{i=n}^{n+\kappa-1} |i\rangle \langle i| \otimes \mathbf{1}_{4\times4}.
$$

These unitary operators with a block structure are known as uniformly controlled quantum gates. They are present in many quantum algorithms and are used to decompose general unitary gates and locally prepare arbitrary quantum states (Möttönen et al., 2004; Mottonen et al., 2004; Bergholm et al., 2005; Arrazola et al., 2022; Park et al., 2019). They are defined as multi-controlled unitaries where each unitary block targets a set of qubits, two qubits in this case, while the other $\log_2(n + \kappa)$ qubits act as control qubits. Multi-controlled unitaries are applied depending on the state of the control qubits, which are unchanged, and only modify the target qubits. These operations generalize the CNOT gate for two qubits, in that we can now have several control and target qubits. Notice that the block structure of the unitary $U_{\boldsymbol{\theta}}$ arises from indexing the targets as the lowest-order bits. Recently, efficient decompositions of multi-controlled unitaries have been proposed in terms of the number of single-qubit and two-qubit gates (Zindorf & Bose, 2024; 2025), as well as for approximations of the multi-controlled gate (Silva et al., 2024). Code implementations of these quantum gates can be found in the *Qclib* library (Araujo et al., 2023). Finally, the identity blocks $\mathbf{1}_{4\times4}$ do not introduce additional gates into the quantum circuit, so the effective circuit can be reduced to the application of the $\bar{U}^{(i)}$ gates. However, the number of control qubits is fixed by $\log_2(n + \kappa)$ and we need all of them to compute the output probabilities, as we will see below.

**Construction of V.** Next, let $V \in \mathbb{C}^{n_U \times n_U}$ be any unitary matrix mapping $|0\rangle^{\otimes \mathfrak{n}}$ to the state $|\psi\rangle = \frac{1}{\sqrt{n}} \sum_{i=0}^{n-1} |4i\rangle$ which, for $n \geq 2$, is also explicitly given as $|\psi\rangle = \frac{1}{\sqrt{n}} \sum_{i=0}^{n-1} |i\rangle \otimes |00\rangle$. Note that different choices of $V$ are possible and the only required property is $V|0\rangle^{\otimes \mathfrak{n}} = |\psi\rangle$. We refer to Appendix D for an example.

**Measuring circuit outputs.** We can now measure the state of the $\mathfrak{n}$-qubit system after applying the gates $V$ and $U$. The possible states that we could measure are given by $0, \ldots, n_U - 1$ (in binary). By running the circuit repeatedly, we can now obtain (up to well-controlled Monte Carlo error, see Appendix E) the probabilities $\mathbb{P}_m^n$ that the measured state is in $\{m, 4+m, \ldots, 4(n-1)+m\}$, for $m \in \{0, 1, 2, 3\}$, where $m$ is the binary state of the last two qubits (the target qubits).

More formally, consider the unitary gate map $C(\boldsymbol{x}, \boldsymbol{z}) = C_{\mathfrak{n},\boldsymbol{\theta}}(\boldsymbol{x}, \boldsymbol{z}) := U_{\boldsymbol{\theta}}(\boldsymbol{x}, \boldsymbol{z})V$ acting on $\mathfrak{n} = 2 + \log_2(n + \kappa)$ qubits. This circuit acts on the initial state $|0\rangle^{\otimes \mathfrak{n}}$ via the quantum gates $V$ and $U$ as

$$
C_{\mathfrak{n},\boldsymbol{\theta}}(\boldsymbol{x}, \boldsymbol{z}) |0\rangle^{\otimes \mathfrak{n}} = \frac{1}{\sqrt{n}} \sum_{i=0}^{n-1} |i\rangle \otimes U_1^{(i+1)}(\boldsymbol{x},\boldsymbol{z}) |0\rangle \otimes U_2^{(i+1)} |0\rangle.
$$

Then, we measure

$$
\mathbb{P}_m^{n,\boldsymbol{\theta}} = \mathbb{P}_m^{n,\boldsymbol{\theta}}(\boldsymbol{x}, \boldsymbol{z}) := \mathbb{P}\left( \text{"} C_{\mathfrak{n},\boldsymbol{\theta}}(\boldsymbol{x}, \boldsymbol{z}) |0\rangle^{\otimes \mathfrak{n}} \in \{m, 4+m, \ldots, 4(n-1)+m\} \text{"} \right). \tag{2}
$$

This is the sum of the probabilities of being in the states $|i\rangle \otimes |m\rangle$, where $i = 0, \ldots, n-1$. That is,

$$
\mathbb{P}_m^{n,\boldsymbol{\theta}}(\boldsymbol{x}, \boldsymbol{z}) = \frac{1}{n} \sum_{i=1}^{n} \left| \langle m| \left( U_1^{(i)}(\boldsymbol{x},\boldsymbol{z}) |0\rangle \otimes U_2^{(i)} |0\rangle \right) \right|^2.
$$

**Parallel circuits.** With $n$ (or equivalently $\mathfrak{n}$) fixed, the quantum circuit introduced above is uniquely defined by the choice of circuit parameters $\boldsymbol{\theta} \in \Theta$. We will now run $N$ such circuits in parallel, each representing a component of the state map $F$ in (1). Each circuit is described by its parameters $\boldsymbol{\theta}^j \in \Theta$, $j \in \{1, \ldots, N\}$. The circuit outputs then induce maps $\mathbb{P}_m^{n,\boldsymbol{\theta}^j} : \mathbb{R}^N \times \mathbb{R}^d \to [0,1]$ by the circuit output probabilities (2) with parameters for the $j$-th circuit given by $\boldsymbol{\theta} = \boldsymbol{\theta}^j$.

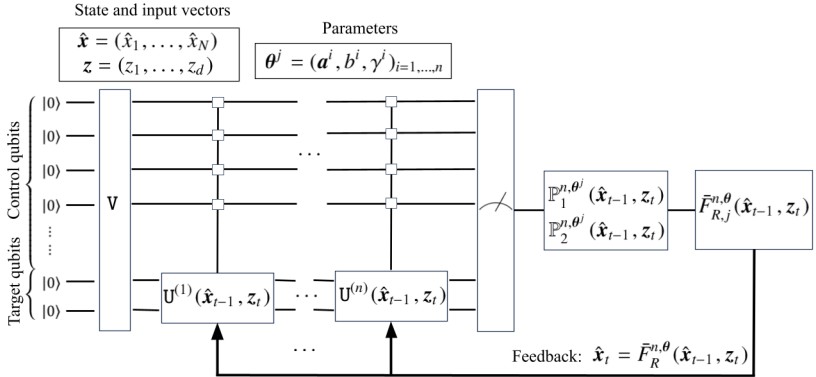

Figure 1: Schematic representation of the $j$-th circuit given by parameters $\boldsymbol{\theta}^j \in \Theta$, $j \in \{1, \ldots, N\}$.

**Recurrent quantum neural networks (RQNN).** With these ingredients, we can now define the RQNN that we will consider. Given the gate map $\mathtt{C}_{\mathfrak{n},\boldsymbol{\theta}}$ and $R > 0$, we define $\bar{F}_R^{n,\boldsymbol{\theta}} : \mathbb{R}^N \times \mathbb{R}^d \to \mathbb{R}^N$ by its component maps $\bar{F}_R^{n,\boldsymbol{\theta}} = (\bar{F}_{R,1}^{n,\boldsymbol{\theta}}, \ldots, \bar{F}_{R,N}^{n,\boldsymbol{\theta}})$. For $j = 1, \ldots, N$, the $j$-th component map $\bar{F}_{R,j}^{n,\boldsymbol{\theta}} : \mathbb{R}^N \times \mathbb{R}^d \to \mathbb{R}$ is defined by

$$\bar{F}_{R,j}^{n,\boldsymbol{\theta}}(\boldsymbol{x}, \boldsymbol{z}) := R - 2R[\mathbb{P}_1^{n,\boldsymbol{\theta}^j}(\boldsymbol{x}, \boldsymbol{z}) + \mathbb{P}_2^{n,\boldsymbol{\theta}^j}(\boldsymbol{x}, \boldsymbol{z})], \quad (\boldsymbol{x}, \boldsymbol{z}) \in \mathbb{R}^N \times \mathbb{R}^d, \tag{3}$$

with $\boldsymbol{\theta} = (\boldsymbol{\theta}^1, \ldots, \boldsymbol{\theta}^N) \in \Theta^N$. Our ***recurrent quantum neural network (RQNN)*** is then defined by the state-space system associated to the state map $\bar{F}_R^{n,\boldsymbol{\theta}}$

$$\hat{\boldsymbol{x}}_t = \bar{F}_R^{n,\boldsymbol{\theta}}(\hat{\boldsymbol{x}}_{t-1}, \boldsymbol{z}_t), \quad t \in \mathbb{Z}_-. \tag{4}$$

Figure 1 provides a schematic representation of how the RQNN acts at each time step for the $j$-th circuit: at any time $t$, the system is initialized, the gates $\mathtt{V}$ and $\mathtt{U}_{\boldsymbol{\theta}^j}(\hat{\boldsymbol{x}}_{t-1}, \boldsymbol{z}_t)$ are applied, and the system is measured. This process is repeated to estimate the probabilities $\mathbb{P}_1^{n,\boldsymbol{\theta}^j}(\hat{\boldsymbol{x}}_{t-1}, \boldsymbol{z}_t)$ and $\mathbb{P}_2^{n,\boldsymbol{\theta}^j}(\hat{\boldsymbol{x}}_{t-1}, \boldsymbol{z}_t)$, which are aggregated into the network output $\bar{F}_{R,j}^{n,\boldsymbol{\theta}}(\hat{\boldsymbol{x}}_{t-1}, \boldsymbol{z}_t)$ according to (3). Once this is done for all $j \in \{1, \ldots, N\}$, the network state $\hat{\boldsymbol{x}}_t$ is stored to be used as feedback for the next time step $t+1$.

In the next paragraphs, we aim to address the following questions:

- Can we choose the parameters $\boldsymbol{\theta}$ in such a way that the system determined by (4) satisfies the echo state property?
- Can the family of systems determined by equations of the type (4) approximate general state-space systems arbitrarily well? More specifically, given an arbitrary state-space map $\boldsymbol{x}_t = F(\boldsymbol{x}_{t-1}, \boldsymbol{z}_t)$ with $F \colon \mathbb{R}^N \times \mathbb{R}^d \to \mathbb{R}^N$ as general as possible, can it be approximated by equations of the type (4)?

## 4 RECURRENT QUANTUM NEURAL NETWORK UNIVERSALITY

This section contains approximation guarantees and universality results for the recurrent quantum neural network (RQNN) family. To achieve this, in Section 4.1 we first prove refined approximation error bounds (that generalize those in Gonon & Jacquier (2025)) for feedforward quantum

neural networks (QNNs) that allow us to control the error committed when approximating a function and its derivatives simultaneously, a crucial ingredient for analysing the RQNN feedback loop. These error bounds show how recurrent QNNs can be used to approximate state-space maps $F$ arbitrarily well as long as these are sufficiently smooth and satisfy Barron-type integrability conditions like, for example, $\int_{\mathbb{R}^N \times \mathbb{R}^d} \xi_i^2 |\widehat{F_j}(\boldsymbol{\xi})| \mathrm{d}\boldsymbol{\xi} < \infty$, for $i = 1, \ldots, N + d$ and $j = 1, \ldots, N$, or $I_q = \int_{\mathbb{R}^N \times \mathbb{R}^d} \|\boldsymbol{\xi}\|^q |\widehat{F_j}(\boldsymbol{\xi})| \mathrm{d}\boldsymbol{\xi} < \infty$, for some $q \geq 2$ (see Proposition 4.2 and Corollary 4.3 below); RQNN state maps are hence universal in that category. These bounds are devised with respect to $L^\infty$ and $L^2$-type norms. As we shall prove, in the $L^\infty$ case, the universality of the RQNN family still holds with respect to state maps that do not necessarily satisfy the Barron condition, even though in that framework we do not formulate approximation bounds. Finally, in the last two sections, we exploit all these results on the approximation of state maps to obtain universality statements and error bounds for the approximation of arbitrary causal, time-invariant, and fading memory filters using a modified recurrent QNN. In addition to the tools developed here, our proofs of these results rely on techniques from Gonon & Ortega (2020; 2021) and the overall strategy is reminiscent of the so-called internal approximation approach introduced in Grigoryeva & Ortega (2018b, Theorem 3.1 (iii)) for echo state networks, which consists of obtaining approximation results for filters out of statements of that type for the state maps that generate them.

The approximation rate in all our results is free from the curse of dimensionality: the error decays as $\frac{1}{\sqrt{n}}$ as we increase $n$, with this rate of decay being independent of the input dimension $d$ and the state space dimension $N$. Moreover, the required number of qubits $\mathfrak{n} = \lceil \log_2(2n) \rceil$ is only growing logarithmically in the accuracy parameter $n$. Put differently, our circuit requires only $\mathcal{O}(\varepsilon^{-2})$ weights and $\mathcal{O}(\lceil \log_2(\varepsilon^{-1}) \rceil)$ qubits suffices to achieve approximation error $\varepsilon > 0$ when approximating functions with sufficiently integrable Fourier transforms.

## 4.1 RQNN APPROXIMATION OF STATE-SPACE MAPS AND THEIR DERIVATIVES

As a first step, we aim to establish RQNN approximation results for a function jointly with its derivatives. Denote by $\mathcal{F}_R$ the class of integrable functions $f : \mathbb{R}^N \times \mathbb{R}^d \to \mathbb{R}$ with Fourier integral bounded above by a constant $R > 0$, that is,

$$\mathcal{F} := \left\{ f : \mathbb{R}^N \times \mathbb{R}^d \to \mathbb{R} : f \in \mathcal{C}\left(\mathbb{R}^N \times \mathbb{R}^d\right) \cap L^1\left(\mathbb{R}^N \times \mathbb{R}^d\right), \quad \|\widehat{f}\|_1 < \infty \right\}, \quad (5)$$

$$\mathcal{F}_R := \left\{ f \in \mathcal{F}, \text{ with } \|\widehat{f}\|_1 \leq R \right\}, \qquad \text{for } R > 0. \quad (6)$$

Here, for a continuous and integrable function $f : \mathbb{R}^N \times \mathbb{R}^d \to \mathbb{R}$ we denote its Fourier transform by $\widehat{f}(\boldsymbol{\xi}_1, \boldsymbol{\xi}_2) := \int_{\mathbb{R}^N \times \mathbb{R}^d} e^{-2\pi \mathrm{i}(\boldsymbol{y}_1, \boldsymbol{y}_2) \cdot (\boldsymbol{\xi}_1, \boldsymbol{\xi}_2)} f(\boldsymbol{y}_1, \boldsymbol{y}_2) \mathrm{d}\boldsymbol{y}_1 \mathrm{d}\boldsymbol{y}_2$, with $(\boldsymbol{\xi}_1, \boldsymbol{\xi}_2) \in \mathbb{R}^N \times \mathbb{R}^d$.

Our first result derives a representation for the QRNN output.

**Proposition 4.1.** *For any* $n \in \mathbb{N}$, $j = 1, \ldots, N$, $\boldsymbol{\theta} = (\boldsymbol{\theta}^1, \ldots, \boldsymbol{\theta}^N) \in \boldsymbol{\Theta}^N$ *with* $\boldsymbol{\theta}^j = (\boldsymbol{a}^{i,j}, b^{i,j}, \gamma^{i,j})_{i=1,\ldots,n} \in \boldsymbol{\Theta}$, *the RQNN introduced in* (3) *can be represented as*

$$\bar{F}_{R,j}^{n,\boldsymbol{\theta}}(\boldsymbol{x}, \boldsymbol{z}) = \frac{1}{n} \sum_{i=1}^n R \cos\left(\gamma^{i,j}\right) \cos\left(b^{i,j} + \boldsymbol{a}^{i,j} \cdot (\boldsymbol{x}, \boldsymbol{z})\right), \quad \text{for all } (\boldsymbol{x}, \boldsymbol{z}) \in \mathbb{R}^N \times \mathbb{R}^d. \quad (7)$$

Our next result provides an approximation error bound for the QRNN state map jointly with its derivatives. The proof is provided in Appendix B.2. Let $\mu$ be an arbitrary probability measure on $(\mathbb{R}^N \times \mathbb{R}^d, \mathcal{B}(\mathbb{R}^N \times \mathbb{R}^d))$. Recall the notation

$$\|f - g\|_{L^2(\mu)} := \left( \int_{\mathbb{R}^N \times \mathbb{R}^d} |f(\boldsymbol{x}, \boldsymbol{z}) - g(\boldsymbol{x}, \boldsymbol{z})|^2 \mu(\mathrm{d}\boldsymbol{x}, \mathrm{d}\boldsymbol{z}) \right)^{1/2}.$$

**Proposition 4.2.** *Let* $R > 0$ *and suppose* $F = (F_1, \ldots, F_N) : \mathbb{R}^N \times \mathbb{R}^d \to \mathbb{R}^N$ *is continuously differentiable and satisfies* $F_j \in \mathcal{F}_R$ *and* $\partial_i F_j \in \mathcal{F}$ *and* $\int_{\mathbb{R}^N \times \mathbb{R}^d} \xi_i^2 |\widehat{F_j}(\boldsymbol{\xi})| \mathrm{d}\boldsymbol{\xi} < \infty$ *for* $i = 1, \ldots, N + d$ *and* $j = 1, \ldots, N$. *Then, for any* $n \in \mathbb{N}$, *there exists* $\boldsymbol{\theta} \in \boldsymbol{\Theta}^N$ *such that*

$$\left\| \bar{F}_{R,j}^{n,\boldsymbol{\theta}} - F_j \right\|_{L^2(\mu)}^2 + \sum_{i=1}^{N+d} \left\| \partial_i \bar{F}_{R,j}^{n,\boldsymbol{\theta}} - \partial_i F_j \right\|_{L^2(\mu)}^2 \leq \frac{C_j}{n},$$

*for any* $j \in \{1, \ldots, N\}$, *where* $C_j = \|\widehat{F_j}\|_1^2 + 4\pi^2 \|\widehat{F_j}\|_1 \int_{\mathbb{R}^N \times \mathbb{R}^d} \sum_{i=1}^{N+d} \xi_i^2 |\widehat{F_j}(\boldsymbol{\xi})| \mathrm{d}\boldsymbol{\xi}$.

Next, we show that it is also possible to also obtain approximation results for QNNs with bounded network coefficients. The proof is provided in Appendix B.3.

**Corollary 4.3.** *In the setting of Proposition 4.2, assume, in addition, that $\int_{\mathbb{R}^N \times \mathbb{R}^d} \|\boldsymbol{\xi}\|^q |\widehat{F_j}(\boldsymbol{\xi})| \mathrm{d}\boldsymbol{\xi} < \infty$ for some $q \geq 2$. Then, for any $n \in \mathbb{N}$, there exists $\boldsymbol{\theta} \in \Theta$ such that for any $j \in \{1, \ldots, N\}$,*

$$\left\| \bar{F}_{R,j}^{n,\boldsymbol{\theta}} - F_j \right\|_{L^2(\mu)}^2 + \sum_{i=1}^{N+d} \left\| \partial_i \bar{F}_{R,j}^{n,\boldsymbol{\theta}} - \partial_i F_j \right\|_{L^2(\mu)}^2 \leq \frac{\bar{C}_j}{n},$$

*where $\bar{C}_j = 3C_j$. Moreover, we can choose $\boldsymbol{\theta} = (\boldsymbol{\theta}^1, \ldots, \boldsymbol{\theta}^N) \in \Theta^N$ with $\boldsymbol{\theta}^j = (\boldsymbol{a}^{i,j}, b^{i,j}, \gamma^{i,j})_{i=1,\ldots,n}$ in such a way that for all $i = 1, \ldots, n$, $j = 1 \ldots, N$,*

$$\|\boldsymbol{a}^{i,j}\| \leq 2\pi \left( 3n \|\widehat{F_j}\|_1^{-1} \int_{\mathbb{R}^N \times \mathbb{R}^d} \|\boldsymbol{\xi}\|^q |\widehat{F_j}(\boldsymbol{\xi})| \mathrm{d}\boldsymbol{\xi} \right)^{\frac{1}{q}}. \tag{8}$$

Next, we complement the $L^2(\mathbb{R}^N \times \mathbb{R}^d, \mu)$-error bound in Proposition 4.2 with a uniform error bound on compact sets. For $M > 0$ and $f, g \in \mathcal{C}(\mathbb{R}^N \times \mathbb{R}^d)$ denote

$$\|f - g\|_{\infty, M} := \sup_{(\boldsymbol{x}, \boldsymbol{z}) \in [-M, M]^N \times [-M, M]^d} |f(\boldsymbol{x}, \boldsymbol{z}) - g(\boldsymbol{x}, \boldsymbol{z})|.$$

**Proposition 4.4.** *Let $R, M > 0$ and suppose $F = (F_1, \ldots, F_N)$ is continuously differentiable and satisfies $F_j \in \mathcal{F}_R$ and $\partial_i F_j \in \mathcal{F}$ and $\int_{\mathbb{R}^N \times \mathbb{R}^d} \|\boldsymbol{\xi}\|^4 |\widehat{F_j}(\boldsymbol{\xi})| \mathrm{d}\boldsymbol{\xi} < \infty$ for $j = 1, \ldots, N$. Then, for any $n \in \mathbb{N}$, there exists $\boldsymbol{\theta} \in \Theta$ such that for any $j \in \{1, \ldots, N\}$,*

$$\left\| \bar{F}_{R,j}^{n,\boldsymbol{\theta}} - F_j \right\|_{\infty, M} + \sum_{i=1}^{N+d} \left\| \partial_i \bar{F}_{R,j}^{n,\boldsymbol{\theta}} - \partial_i F_j \right\|_{\infty, M} \leq \frac{C_j^\infty}{\sqrt{n}}, \tag{9}$$

*where $C_j^\infty = 2(\pi + 1) \|\widehat{F_j}\|_1 + (8\pi M + 4\pi^2)(N+d)^{\frac{1}{2}} \|\widehat{F_j}\|_1^{\frac{1}{2}} I_{2,j}^{1/2} + 16M\pi^2(N+d) \|\widehat{F_j}\|_1^{1/2} I_{4,j}^{1/2}$ for $I_{q,j} = \int_{\mathbb{R}^N \times \mathbb{R}^d} \|\boldsymbol{\xi}\|^q |\widehat{F_j}(\boldsymbol{\xi})| \mathrm{d}\boldsymbol{\xi} < \infty$.*

The proof can be found in Appendix B.4. Finally, we obtain a qualitative universal approximation result for QNNs jointly with their derivatives. The proof can be found in Appendix B.5.

**Corollary 4.5.** *Let $F = (F_1, \ldots, F_N)$ be continuously differentiable. Then for any $\varepsilon > 0$ and $\mathcal{X} \subset \mathbb{R}^N \times \mathbb{R}^d$ compact there exist $n \in \mathbb{N}$, $R > 0$ and $\boldsymbol{\theta} \in \Theta$ such that for any $j \in \{1, \ldots, N\}$, $\bar{F}_{R,j}^{n,\boldsymbol{\theta}}$ satisfies*

$$\sup_{(\boldsymbol{x}, \boldsymbol{z}) \in \mathcal{X}} |F_j(\boldsymbol{x}, \boldsymbol{z}) - \bar{F}_{R,j}^{n,\boldsymbol{\theta}}(\boldsymbol{x}, \boldsymbol{z})| + \|\nabla F_j(\boldsymbol{x}, \boldsymbol{z}) - \nabla \bar{F}_{R,j}^{n,\boldsymbol{\theta}}(\boldsymbol{x}, \boldsymbol{z})\| \leq \varepsilon. \tag{10}$$

## 4.2 RECURRENT QNN APPROXIMATION BOUNDS FOR STATE-SPACE FILTERS

The results in the previous section show that the family of RQNNs that were introduced in (3) is capable of approximating arbitrarily well the very general class of continuously differentiable state-space maps with bounded Fourier transform, together with their derivatives. These approximations hold with respect to both the $L^2$ norm (Proposition 4.2 and Corollary 4.3) and the $L^\infty$ norm on compacta (Proposition 4.4 and Corollary 4.5). We will now use the uniform RQNN approximation results for the state maps to conclude similar uniform approximation results for the corresponding filters under additional hypotheses that guarantee that those exist.

Consider a state-space system

$$\boldsymbol{x}_t = F(\boldsymbol{x}_{t-1}, \boldsymbol{z}_t), \quad t \in \mathbb{Z}_-, \tag{11}$$

with state process $(\boldsymbol{x}_t)_{t \in \mathbb{Z}_-}$ valued in $\mathbb{R}^N$, input process $(\boldsymbol{z}_t)_{t \in \mathbb{Z}_-}$ valued in $\mathbb{R}^d$ and $F \colon \mathbb{R}^N \times \mathbb{R}^d \to \mathbb{R}^N$. We work under the assumption that $F$ is contractive and satisfies Barron-type integrability conditions (Barron, 1992; 1993; Barron & Klusowski, 2018). Then, e.g., Proposition 1 and Remark 2 in Gonon et al. (2020) imply that, for any compact $D_d \subset \mathbb{R}^d$, the associated filter $U^F \colon (D_d)^{\mathbb{Z}_-} \to (B_N)^{\mathbb{Z}_-}$ induced by the restriction of $F$ to $B_N \times D_d$ is well-defined and continuous.

Our next result shows that among the RQNNs that we discussed in Proposition 4.4 there exist systems that have the echo state property and hence have a filter associated. More importantly, those filters can be used to uniformly approximate any of the filters corresponding to the general systems introduced above in (11) as long as they satisfy a Barron-type integrability condition and are sufficiently contractive. The proof can be found in Appendix C.1. Here, $\| \cdot \|_2$ is the spectral norm. In particular, this result shows that the error rate is free from the curse of dimensionality: the error decays as $\frac{1}{\sqrt{n}}$ as we increase $n$, with this rate of decay being independent of the input dimension $d$ and the state space dimension $N$. Thus, the RQNN requires only $\mathcal{O}(\varepsilon^{-2})$ weights and $\mathcal{O}(\lceil \log_2(\varepsilon^{-1}) \rceil)$ qubits to achieve approximation error $\varepsilon > 0$ for the considered state-space systems.

**Theorem 4.6.** *Suppose $F$ in (11) is continuously differentiable with $\|\nabla_{\boldsymbol{x}} F(\boldsymbol{x}, \boldsymbol{z})\|_2 \leq \lambda$ for all $\boldsymbol{x} \in \mathbb{R}^N, \boldsymbol{z} \in D_d$ for some $\lambda \in (0, 1)$ and, moreover, $F$ satisfies $F_j \in \mathcal{F}_R$, $\partial_i F_j \in \mathcal{F}$ and $\int_{\mathbb{R}^N \times \mathbb{R}^d} \|\boldsymbol{\xi}\|^4 |\widehat{F_j}(\boldsymbol{\xi})| \mathrm{d}\boldsymbol{\xi} < \infty$ for $j = 1, \ldots, N$. Denote by $U^F : (D_d)^{\mathbb{Z}_-} \to (B_N)^{\mathbb{Z}_-}$ the filter associated to (11). Then for any $n \in \mathbb{N}$ with $n > n_0$ there exists $\boldsymbol{\theta} \in \boldsymbol{\Theta}$ such that the system (4) has the echo state property and the associated filter $\bar{U} : (D_d)^{\mathbb{Z}_-} \to (\mathbb{R}^N)^{\mathbb{Z}_-}$ satisfies*

$$\sup_{\boldsymbol{z} \in (D_d)^{\mathbb{Z}_-}} \sup_{t \in \mathbb{Z}_-} \|U^F(\boldsymbol{z})_t - \bar{U}(\boldsymbol{z})_t\| \leq \frac{1}{1-\lambda} \frac{\sqrt{N} \max_{j=1,\ldots,N} C_j^{\infty}}{\sqrt{n}}. \tag{12}$$

*Here, $n_0$ may be chosen as $n_0 = N^2 \frac{(\max_{j=1,\ldots,N} C_j^{\infty})^2}{(1-\lambda)^2}$.*

Notice that $N$ represents the state space dimension of the target $F$, which is matched by the QRNN dimension to obtain the approximation error bound. Theorem 4.6 also proves an advantage of QRNNs over classical RNNs. RNN approximation bounds for state-space systems driven by Barron-type functions were obtained in (Gonon et al., 2023, Theorem 3). While the approximation rate in Theorem 4.6 is the same ($\frac{1}{2}$ in both cases), the Fourier integrability condition required in the quantum case is *strictly weaker*. Specifically, the condition $\int_{\mathbb{R}^N \times \mathbb{R}^d} \|\boldsymbol{\xi}\|^4 |\widehat{F_j}(\boldsymbol{\xi})| \mathrm{d}\boldsymbol{\xi} < \infty$ implies that the smoothness assumption (Gonon et al., 2023, Definition 2) required for (Gonon et al., 2023, Theorem 3) is satisfied. For example, consider a Sobolev function $F \in H^s(\mathbb{R}^N \times \mathbb{R}^d)$. Then, the integrability condition for the QRNN approximation result is satisfied for any $s > \frac{N+d}{2} + 4$ (by (Folland, 2020, Lemma 6.5) and its proof). In contrast, the integrability condition for the RNN approximation result in (Gonon et al., 2023, Theorem 3) would require the stronger condition $s > N + d + 3$.

### 4.3 UNIVERSALITY

In the previous section, we proved error bounds for the approximation using recurrent QNNs of the filters induced by contractive state-space targets with Barron-type integrability conditions. These bounds show, in passing, the universality of the family of RQNN filters in that category. We now extend this universality statement (without formulating error bounds) to the much larger family of fading memory filters by introducing a modification in the RQNN reservoir. We define $\tilde{F}_R^{n,\boldsymbol{\theta}} : \mathbb{R}^N \times \mathbb{R}^d \to \mathbb{R}^N$ by its component maps $\tilde{F}_R^{n,\boldsymbol{\theta}} = (\tilde{F}_{R,1}^{n,\boldsymbol{\theta}}, \ldots, \tilde{F}_{R,N}^{n,\boldsymbol{\theta}})$. For $j = 1, \ldots, N$, the $j$-th component map $\tilde{F}_{R,j}^{n,\boldsymbol{\theta}} : \mathbb{R}^N \times \mathbb{R}^d \to \mathbb{R}$ is defined by

$$\tilde{F}_{R,j}^{n,\boldsymbol{\theta}}(\boldsymbol{x}, \boldsymbol{z}) := R - 2R[\mathbb{P}_1^{n,\boldsymbol{\theta}^j}(P_j \boldsymbol{x}, \boldsymbol{z}) + \mathbb{P}_2^{n,\boldsymbol{\theta}^j}(P_j \boldsymbol{x}, \boldsymbol{z})], \quad (\boldsymbol{x}, \boldsymbol{z}) \in \mathbb{R}^N \times \mathbb{R}^d, \tag{13}$$

with $\boldsymbol{\theta} = (\boldsymbol{\theta}^1, \ldots, \boldsymbol{\theta}^N) \in \boldsymbol{\Theta}^N$ and $P_1, \ldots, P_N \in \mathbb{R}^{N \times N}$ linear preprocessing maps. Our modified RQNN is then defined by the state-space system associated to the state map $\tilde{F}_R^{n,\boldsymbol{\theta}}$

$$\hat{\boldsymbol{x}}_t = \tilde{F}_R^{n,\boldsymbol{\theta}}(\hat{\boldsymbol{x}}_{t-1}, \boldsymbol{z}_t), \quad t \in \mathbb{Z}_-. \tag{14}$$

The next lemma (with proof provided in Appendix C.2) shows that adding linear preprocessing maps to reservoir equations can lead to the echo state property without contraction assumptions.

**Lemma 4.7.** *Let $\tilde{F} = (\tilde{F}_1, \ldots, \tilde{F}_N)$ be a reservoir map where each component $\tilde{F}_j : \mathbb{R}^N \times \mathbb{R}^d \to \mathbb{R}$, for $j = 1, \ldots, N$, is defined as*

$$\tilde{F}_j(\boldsymbol{x}, \boldsymbol{z}) = g_j(P_j \boldsymbol{x}, \boldsymbol{z}) \tag{15}$$

*where $P_1, \ldots, P_N \in \mathbb{R}^{N \times N}$ are linear preprocessing maps for any maps $g_j : \mathbb{R}^N \times \mathbb{R}^d \to \mathbb{R}$, $j = 1, \ldots, N$. Define an arbitrary partition of the state vector $\hat{\boldsymbol{x}}_t = [\hat{\boldsymbol{x}}_t^{(1)}, \ldots, \hat{\boldsymbol{x}}_t^{(K)}] \in \mathbb{R}^{I_1} \times \cdots \times \mathbb{R}^{I_K}$*

*such that $\sum_{k=1}^{K} I_k = N > 0$ and $I_k \geq 1$ for all $t \in \mathbb{Z}_-$. We define the index $l_k = \sum_{s=1}^{k} I_s$ for $k = 1, \ldots, K$. For $k = 1, j \in \{1, \ldots, l_1\}$, and $k = 2, \ldots, K-1, j \in \{l_{k-1}+1, \ldots, l_k\}$, select $P_j$ as the matrix with zero entries, except for $(P_j)_{l,l+l_k} = 1$ for $l = 1, \ldots, \sum_{s=k+1}^{K} I_s$ and let $P_j = 0$ for $j = l_{K-1}+1, \ldots, N$. Then, the map $\tilde{F}$ has the echo state property for any $N \in \mathbb{N}^+$.*

Notice that Lemma 4.7 provides the echo state property by imposing a finite memory of $K-1$ time steps on the reservoir. Let $D_d \subset \mathbb{R}^d$, $B_m \subset \mathbb{R}^m$ be compact. For a readout $W \in \mathbb{R}^{m \times N}$, denote

$$\mathbf{y}_t = W \boldsymbol{x}_t \tag{16}$$

the output process associated to the recurrent QNNs (4) and (14). Our next result proves universality of RQNNs. The proof is provided in Appendix C.3.

**Theorem 4.8.** *Let $U : (D_d)^{\mathbb{Z}_-} \to (B_m)^{\mathbb{Z}_-}$ be a causal and time-invariant filter that satisfies the fading memory property (that is, it is continuous with respect to the product topology). Then, for any $\varepsilon > 0$ there exist $n, N \in \mathbb{N}$, preprocessing matrices $P_1, \ldots, P_N \in \mathbb{R}^{N \times N}$, a readout $W \in \mathbb{R}^{m \times N}$, and circuit parameters $\boldsymbol{\theta} \in \boldsymbol{\Theta}^N$ such that the RQNN (14) has the echo state property and the filter $\bar{U}_W : (D_d)^{\mathbb{Z}_-} \to (B_m)^{\mathbb{Z}_-}$ associated to the output process (16) satisfies*

$$\sup_{\boldsymbol{z} \in (D_d)^{\mathbb{Z}_-}} \sup_{t \in \mathbb{Z}_-} \left\| U(\boldsymbol{z})_t - \bar{U}_W(\boldsymbol{z})_t \right\| \leq \varepsilon. \tag{17}$$

## 5 CONCLUSIONS

Approximation bounds and universality properties are part of the theoretical cornerstone of machine learning models. While some studies have addressed the question of universality for QRC models, the combination of the two had not previously been explored in the context of recurrent QNNs. In this paper, we derived approximation bounds and universality statements for recurrent QNNs based on the circuit implementation presented in Gonon & Jacquier (2025), which is compatible with hardware deployment and whose implementation with Rydberg atoms has been already discussed in Agarwal et al. (2024). This circuit uses a uniformly controlled quantum gate to apply multi-controlled rotations to a set of control and target qubits, and it has been recently shown that it can be efficiently implemented (Zindorf & Bose, 2024; Silva et al., 2024; Zindorf & Bose, 2025).

To prove our results, we first derived approximation bounds for the static version of the QNN and its derivatives. These results are used in Theorem 4.6 to provide filter approximation bounds that show that RQNNs are able to uniformly approximate the filters induced by any contracting Barron-type state-space system. Finally, Theorem 4.8 extends this universality property to the much larger category of arbitrary fading memory, causal, and time-invariant filters. In this last result, neither Barron-type integrability nor contractivity conditions are needed for the target filter. While our results apply to variational systems in which all parameters are trainable, they pave the way for results on quantum reservoir systems in which some parameters in the recurrent layer are randomly generated and only the output layer weights are tuned. Which strategy is best in terms of speed and accuracy will depend on the number of blocks $n$ of the circuit, the intrinsic noise of the hardware, and the target task. Future research will focus on implementing and comparing the variational and reservoir approaches.

This work paves the way for extending the theoretical analysis of QRC models beyond the state-affine system (SAS) paradigm (Martínez-Peña & Ortega, 2023). It is important to understand in which situations the feedback approach is preferable to other protocols. Questions such as the exponential concentration of observables (Sannia et al., 2025; Xiong et al., 2025) and the suitability of QRC models for learning quantum temporal tasks (Tran & Nakajima, 2021; Nokkala, 2023) are fundamental to discerning the conditions that render QRC models more useful than classical machine learning approaches.

While our paper obtains approximation bounds for Barron-type sate-space systems, an important direction of future research will consist in studying approximation error rates for systems with high degrees of roughness or non-contractive dynamics. Furthermore, our paper focuses on approximation properties of RQNNs. Gradient-based training approaches for optimizing RQNN parameters have been proposed, e.g., in Bausch (2020); Li et al. (2023); Siemaszko et al. (2023). Quantum circuit training may face *Barren plateaus* McClean et al. (2018); Larocca et al. (2025), flat parameter optimization landscapes for large number of qubits. Developing efficient training algorithms and studying these effects in detail will be a further important direction for future research.

ACKNOWLEDGMENTS

The authors acknowledge partial financial support from the School of Physical and Mathematical Sciences of the Nanyang Technological University through the SPMS Collaborative Research Award 2023 entitled "Quantum Reservoir Systems for Machine Learning". RMP acknowledges the QCDI project funded by the Spanish Government. JPO wishes to thank the hospitality of the Donostia International Physics Center and LG and RMP that of the Division of Mathematical Sciences of the Nanyang Technological University, during the academic visits in which some of this work was developed.

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

## APPENDIX

## A QUANTUM RESERVOIR COMPUTING PROTOCOLS

For learning problems with temporal structure, quantum reservoir computing (QRC) has emerged as a promising approach for exploiting noisy intermediate-scale quantum (NISQ) technologies. These include ion traps, nuclear magnetic resonance, cold atoms, photonic platforms, and superconducting qubits (Mujal et al., 2021). When implementing QRC models experimentally, it is necessary to consider the backaction and statistical effects introduced by quantum measurements. Backaction refers to the modification of a quantum state after monitoring, also known as wavefunction collapse. Due to the probabilistic nature of quantum theory, measurements must be repeated to compute the expected values of observables, which introduces a statistical component in all these methodologies. Most available experimental implementations rely on the quantum computer paradigm (Dasgupta et al., 2022; Mlika et al., 2023; Suzuki et al., 2022; Yasuda et al., 2023; Chen et al., 2020; Kubota et al., 2023; Molteni et al., 2023; Pfeffer et al., 2022; Ahmed et al., 2025; Hu et al., 2024; Miranda & Shaji, 2025). However, there is an increasing interest in extending this technique to new settings, such as optical pulses (García-Beni et al., 2023; Paparelle et al., 2025), Rydberg atoms (Bravo et al., 2022; Kornjača et al., 2024), and quantum memristors (Spagnolo et al., 2022; Selimović et al., 2025).

Early QRC model implementations relied on the simplest possible approach, namely, the restarting protocol (Dasgupta et al., 2022; Suzuki et al., 2022; Kubota et al., 2023; Chen et al., 2020; Molteni et al., 2023). In this approach, the expected values of observables are obtained by rerunning the algorithm from the first time step at each subsequent time step. This avoids the backaction effect of quantum measurements. However, the complexity of this protocol scales quadratically with the length of the input sequence, making it very time-consuming. A faster alternative is the rewinding protocol (Mujal et al., 2021; Čindrak et al., 2024), where the fading memory of the quantum reservoir is exploited to restart the algorithm with a fixed window of past time steps. This reduces the complexity of the algorithm to linear in terms of input length. Originally proposed in Chen et al. (2020), this protocol has thus far only been considered numerically (Mujal et al., 2023; Čindrak et al., 2024).

Both the restarting and rewinding protocols use repetition of previous time steps to reproduce the dynamics of the theoretical model and avoid the disruptive effect of projective measurements used to extract output information. This comes at the cost of halting the quantum dynamics at each time step and the need to buffer the input sequence. Consequently, these approaches lack one of the most important features of traditional reservoir computing, namely, the ability to process information in real time.

New protocols have been proposed to circumvent this problem. The online protocol (Mujal et al., 2023; Franceschetto et al., 2024) uses weak measurements to find a balance between erasing and extracting information. Mid-circuit measurements and reset operations (Hu et al., 2024) can split the reservoir into two parts: memory and readout. The memory retains previous inputs, while measurements only affect the readout part. The feedback protocol (Kobayashi et al., 2024), which can be traced back to QRC with quantum memristors (Spagnolo et al., 2022) and hybrid QRC techniques (Pfeffer et al., 2022; 2023), reinjects the measured observables at each time step as parameters of an input quantum channel. This ensures that no backaction effects are present and that past input information is preserved. Note that in order to compute the observables in real time, these protocols all require several copies of the system to be run in parallel. Furthermore, these protocols can be combined with each other. For instance, the feedback protocol has been combined with both the online protocol (Monomi et al., 2025) and with mid-circuit measurements and reset operations (Murauer et al., 2025).

Of all these approaches, the feedback protocol presents some particularly interesting features. First, the feedback protocol enables us to compute the expected values of observables from a single copy of the system by repeating one time step only. If only a few copies of the system are available, this reduces the experimental time overhead for real-time applications compared to other approaches. Second, in contrast to previous QRC models, where an erasure mechanism is added to provide fundamental properties such as the echo state property, simple unitary operations can provide these properties (Kobayashi et al., 2024). Finally, the dynamical equations of quantum reservoirs under the feedback protocol go beyond the standard state-affine system (SAS) paradigm of QRC models (Martínez-Peña & Ortega, 2023). These properties make the feedback protocol a promising candidate for exploring QRC applications.

## B  PROOFS FOR SECTION 4.1

### B.1  PROOF OF PROPOSITION 4.1

*Proof.* The proof is a modification of the argument used to obtain (Gonon & Jacquier, 2025, Proposition 1). Recall that

$$\bar{F}_{R,j}^{n,\boldsymbol{\theta}}(\boldsymbol{x},\boldsymbol{z}) := R - 2R[\mathbb{P}_1^{n,\boldsymbol{\theta}^j}(\boldsymbol{x},\boldsymbol{z}) + \mathbb{P}_2^{n,\boldsymbol{\theta}^j}(\boldsymbol{x},\boldsymbol{z})], \quad (\boldsymbol{x},\boldsymbol{z}) \in \mathbb{R}^N \times \mathbb{R}^d. \tag{18}$$

Fix $(\boldsymbol{x},\boldsymbol{z}) \in \mathbb{R}^N \times \mathbb{R}^d$ and $j \in \{1, \dots, N\}$ and write $\mathbb{P}_m := \mathbb{P}_m^{n,\boldsymbol{\theta}^j}(\boldsymbol{x},\boldsymbol{z})$ for $m \in \{0, 1, 2, 3\}$. To prove the representation (7), let us first calculate $\mathbb{P}_m$.

As a first step, write

$$\mathtt{UV}\,|0\rangle^{\otimes\mathbf{n}} = \mathtt{U}\,|\psi\rangle = \frac{1}{\sqrt{n}} \sum_{l=0}^{n-1} \mathtt{U}\,|4l\rangle$$

$$= \frac{1}{\sqrt{n}} \sum_{l=0}^{n-1} \sum_{k=0}^{3} \left[\mathtt{U}_1^{(l+1)} \otimes \mathtt{U}_2^{(l+1)}\right]_{k+1,1} |4l+k\rangle .$$

Thus, for $m \in \{0, 1, 2, 3\}$, we have

$$
\begin{aligned}
\mathbb{P}_m &= \sum_{i=0}^{n-1} \left| \langle 4i + m | \, \mathtt{UV} \, |0\rangle^{\otimes \mathtt{n}} \right|^2 \\
&= \sum_{i=0}^{n-1} \left| \langle 4i + m | \, \frac{1}{\sqrt{n}} \sum_{l=0}^{n-1} \sum_{k=0}^{3} \left[ \mathtt{U}_1^{(l+1)} \otimes \mathtt{U}_2^{(l+1)} \right]_{k+1,1} |4l + k\rangle \right|^2 \\
&= \frac{1}{n} \sum_{i=0}^{n-1} \left| \left[ \mathtt{U}_1^{(i+1)} \otimes \mathtt{U}_2^{(i+1)} \right]_{m+1,1} \right|^2 .
\end{aligned}
$$

Next, we may calculate

$$
[\mathtt{U}_1^{(i)} \otimes \mathtt{U}_2^{(i)}]_{1,1} = [\mathtt{U}_1^{(i)}]_{1,1} [\mathtt{U}_2^{(i)}]_{1,1} = \cos\left(\frac{\gamma^{i,j}}{2}\right) \cos\left(\frac{b^{i,j} + \boldsymbol{a}^{i,j} \cdot (\boldsymbol{x}, \boldsymbol{z})}{2}\right),
$$

$$
[\mathtt{U}_1^{(i)} \otimes \mathtt{U}_2^{(i)}]_{2,1} = [\mathtt{U}_1^{(i)}]_{1,1} [\mathtt{U}_2^{(i)}]_{2,1} = \sin\left(\frac{\gamma^{i,j}}{2}\right) \cos\left(\frac{b^{i,j} + \boldsymbol{a}^{i,j} \cdot (\boldsymbol{x}, \boldsymbol{z})}{2}\right),
$$

$$
[\mathtt{U}_1^{(i)} \otimes \mathtt{U}_2^{(i)}]_{3,1} = [\mathtt{U}_1^{(i)}]_{2,1} [\mathtt{U}_2^{(i)}]_{1,1} = \mathrm{i}\cos\left(\frac{\gamma^{i,j}}{2}\right) \sin\left(\frac{b^{i,j} + \boldsymbol{a}^{i,j} \cdot (\boldsymbol{x}, \boldsymbol{z})}{2}\right),
$$

$$
[\mathtt{U}_1^{(i)} \otimes \mathtt{U}_2^{(i)}]_{4,1} = [\mathtt{U}_1^{(i)}]_{2,1} [\mathtt{U}_2^{(i)}]_{2,1} = \mathrm{i}\sin\left(\frac{\gamma^{i,j}}{2}\right) \sin\left(\frac{b^{i,j} + \boldsymbol{a}^{i,j} \cdot (\boldsymbol{x}, \boldsymbol{z})}{2}\right),
$$

and thus

$$
\mathbb{P}_0 = \frac{1}{n} \sum_{i=1}^{n} \cos\left(\frac{\gamma^{i,j}}{2}\right)^2 \cos\left(\frac{b^{i,j} + \boldsymbol{a}^{i,j} \cdot (\boldsymbol{x}, \boldsymbol{z})}{2}\right)^2
$$

$$
\mathbb{P}_1 = \frac{1}{n} \sum_{i=1}^{n} \sin\left(\frac{\gamma^{i,j}}{2}\right)^2 \cos\left(\frac{b^{i,j} + \boldsymbol{a}^{i,j} \cdot (\boldsymbol{x}, \boldsymbol{z})}{2}\right)^2
$$

$$
\mathbb{P}_2 = \frac{1}{n} \sum_{i=1}^{n} \cos\left(\frac{\gamma^{i,j}}{2}\right)^2 \sin\left(\frac{b^{i,j} + \boldsymbol{a}^{i,j} \cdot (\boldsymbol{x}, \boldsymbol{z})}{2}\right)^2
$$

$$
\mathbb{P}_3 = \frac{1}{n} \sum_{i=1}^{n} \sin\left(\frac{\gamma^{i,j}}{2}\right)^2 \sin\left(\frac{b^{i,j} + \boldsymbol{a}^{i,j} \cdot (\boldsymbol{x}, \boldsymbol{z})}{2}\right)^2 .
$$

Therefore, using $\cos(y)^2 = \frac{\cos(2y)+1}{2}$, we obtain

$$
\mathbb{P}_0 + \mathbb{P}_1 = \frac{1}{n} \sum_{i=1}^{n} \cos\left(\frac{b^{i,j} + \boldsymbol{a}^{i,j} \cdot (\boldsymbol{x}, \boldsymbol{z})}{2}\right)^2 = \frac{1}{2} + \frac{1}{2n} \sum_{i=1}^{n} \cos\left(b^{i,j} + \boldsymbol{a}^{i,j} \cdot (\boldsymbol{x}, \boldsymbol{z})\right),
$$

$$
\mathbb{P}_0 + \mathbb{P}_2 = \frac{1}{n} \sum_{i=1}^{n} \cos\left(\frac{\gamma^{i,j}}{2}\right)^2 = \frac{1}{2} + \frac{1}{2n} \sum_{i=1}^{n} \cos\left(\gamma^{i,j}\right).
$$

Putting it all together we obtain, for any given $R > 0$, that

$$
\begin{aligned}
\bar{F}_{R,j}^{n,\boldsymbol{\theta}}(\boldsymbol{x}, \boldsymbol{z}) &= R - 2R[\mathbb{P}_1^{n,\boldsymbol{\theta}^j}(\boldsymbol{x}, \boldsymbol{z}) + \mathbb{P}_2^{n,\boldsymbol{\theta}^j}(\boldsymbol{x}, \boldsymbol{z})] \\
&= R\left[1 + 4\mathbb{P}_0 - 2\left(\mathbb{P}_0 + \mathbb{P}_1\right) - 2\left(\mathbb{P}_0 + \mathbb{P}_2\right)\right] \\
&= \frac{1}{n} \sum_{i=1}^{n} R\cos\left(\gamma^{i,j}\right) \cos\left(b^{i,j} + \boldsymbol{a}^{i,j} \cdot (\boldsymbol{x}, \boldsymbol{z})\right).
\end{aligned}
$$

$\square$

### B.2   PROOF OF PROPOSITION 4.2

*Proof.* Let $j \in \{1, \dots, N\}$ be fixed. As in the proof of Proposition 2 in Gonon & Jacquier (2025), we may use the Fourier inversion theorem to represent

$$
F_j(\boldsymbol{x}, \boldsymbol{z}) = \int_{\mathbb{R}^N \times \mathbb{R}^d} e^{2\pi \mathrm{i}(\boldsymbol{x}, \boldsymbol{z}) \cdot (\boldsymbol{\xi}_1, \boldsymbol{\xi}_2)} \widehat{F_j}(\boldsymbol{\xi}_1, \boldsymbol{\xi}_2) \mathrm{d}\boldsymbol{\xi}_1 \mathrm{d}\boldsymbol{\xi}_2,
$$

which we may rewrite as, with $\boldsymbol{\xi} = (\boldsymbol{\xi}_1, \boldsymbol{\xi}_2)$,

$$F_j(\boldsymbol{x}, \boldsymbol{z}) = \int_{\mathbb{R}^N \times \mathbb{R}^d} \left\{ \cos\left(2\pi(\boldsymbol{x}, \boldsymbol{z}) \cdot \boldsymbol{\xi}\right) \mathrm{Re}[\widehat{F_j}(\boldsymbol{\xi})] + \cos\left(2\pi(\boldsymbol{x}, \boldsymbol{z}) \cdot \boldsymbol{\xi} + \frac{\pi}{2}\right) \mathrm{Im}[\widehat{F_j}(\boldsymbol{\xi})] \right\} d\boldsymbol{\xi} \tag{19}$$

The hypothesis $\partial_i F_j \in \mathcal{F}$ implies that $\int_{\mathbb{R}^N \times \mathbb{R}^d} |\xi_i| |\widehat{F_j}(\boldsymbol{\xi})| d\boldsymbol{\xi} < \infty$. Hence, applying differentiation under the integral sign yields

$$\partial_i F_j(\boldsymbol{x}, \boldsymbol{z}) = -2\pi \int_{\mathbb{R}^N \times \mathbb{R}^d} \left\{ \xi_i \sin\left(2\pi(\boldsymbol{x}, \boldsymbol{z}) \cdot \boldsymbol{\xi}\right) \mathrm{Re}[\widehat{F_j}(\boldsymbol{\xi})] + \xi_i \sin\left(2\pi(\boldsymbol{x}, \boldsymbol{z}) \cdot \boldsymbol{\xi} + \frac{\pi}{2}\right) \mathrm{Im}[\widehat{F_j}(\boldsymbol{\xi})] \right\} d\boldsymbol{\xi}. \tag{20}$$

Next, consider the random function

$$\Phi_j(\boldsymbol{x}, \boldsymbol{z}) := \frac{1}{n} \sum_{i=1}^{n} W_i \cos(B_i + \mathbf{A}_i \cdot (\boldsymbol{x}, \boldsymbol{z})) \tag{21}$$

for randomly selected weights $W_1, \ldots, W_n$, $B_1, \ldots, B_n$ and $\mathbf{A}_1, \ldots, \mathbf{A}_n$ valued in $\mathbb{R}$, $\mathbb{R}$, and $\mathbb{R}^N \times \mathbb{R}^d$, respectively (for notational simplicity we leave the dependence on $j$ implicit here). The distributions of these random variables are chosen as follows. First, we let $Z_1, \ldots, Z_n$ be i.i.d. Bernoulli random variables with

$$\mathbb{P}(Z_i = 1) = \frac{\int_{\mathbb{R}^N \times \mathbb{R}^d} |\mathrm{Re}[\widehat{F_j}(\boldsymbol{\xi})]| d\boldsymbol{\xi}}{\int_{\mathbb{R}^N \times \mathbb{R}^d} |\widehat{F_j}(\boldsymbol{\xi})| d\boldsymbol{\xi}}, \qquad \mathbb{P}(Z_i = 0) = \frac{\int_{\mathbb{R}^N \times \mathbb{R}^d} |\mathrm{Im}[\widehat{F_j}(\boldsymbol{\xi})]| d\boldsymbol{\xi}}{\int_{\mathbb{R}^N \times \mathbb{R}^d} |\widehat{F_j}(\boldsymbol{\xi})| d\boldsymbol{\xi}}. \tag{22}$$

and let $\nu_{\mathrm{Re}}$ and $\nu_{\mathrm{Im}}$ be the probability measures on $\mathbb{R}^N \times \mathbb{R}^d$ with densities

$$\frac{|\mathrm{Re}[\widehat{F_j}]|}{\int_{\mathbb{R}^N \times \mathbb{R}^d} |\mathrm{Re}[\widehat{F_j}(\boldsymbol{\xi})]| d\boldsymbol{\xi}} \qquad \text{and} \qquad \frac{|\mathrm{Im}[\widehat{F_j}]|}{\int_{\mathbb{R}^N \times \mathbb{R}^d} |\mathrm{Im}[\widehat{F_j}(\boldsymbol{\xi})]| d\boldsymbol{\xi}}, \tag{23}$$

respectively. In case $\int_{\mathbb{R}^N \times \mathbb{R}^d} |\mathrm{Re}[\widehat{F_j}(\boldsymbol{\xi})]| d\boldsymbol{\xi} = 0$, instead we choose for $\nu_{\mathrm{Re}}$ an arbitrary probability measure and analogously for $\nu_{\mathrm{Im}}$ in case $\int_{\mathbb{R}^N \times \mathbb{R}^d} |\mathrm{Im}[\widehat{F_j}(\boldsymbol{\xi})]| d\boldsymbol{\xi} = 0$. Next, let $\mathbf{U}_1^{\mathrm{Re}}, \ldots, \mathbf{U}_n^{\mathrm{Re}}$ (resp. $\mathbf{U}_1^{\mathrm{Im}}, \ldots, \mathbf{U}_n^{\mathrm{Im}}$) be i.i.d. random variables with distribution $\nu_{\mathrm{Re}}$ (resp. $\nu_{\mathrm{Im}}$) and assume that $\mathbf{U}_1^{\mathrm{Im}} \ldots, \mathbf{U}_n^{\mathrm{Im}}, \mathbf{U}_1^{\mathrm{Re}}, \ldots, \mathbf{U}_n^{\mathrm{Re}}, Z_1, \ldots, Z_n$ are independent. With these preparations, we are now ready to define the weights in (21):

$$\mathbf{A}_i := 2\pi(Z_i \mathbf{U}_i^{\mathrm{Re}} + (1 - Z_i) \mathbf{U}_i^{\mathrm{Im}}), \qquad B_i := \frac{\pi}{2}(1 - Z_i),$$

$$W_i := \|\widehat{F_j}\|_1 \left[ \frac{\mathrm{Re}[\widehat{F_j}](\mathbf{U}_i^{\mathrm{Re}})}{|\mathrm{Re}[\widehat{F_j}](\mathbf{U}_i^{\mathrm{Re}})|} Z_i + \frac{\mathrm{Im}[\widehat{F_j}](\mathbf{U}_i^{\mathrm{Im}})}{|\mathrm{Im}[\widehat{F_j}](\mathbf{U}_i^{\mathrm{Im}})|}(1 - Z_i) \right],$$

with the quotient set to zero when the denominator is null.

Our goal now is to estimate

$$\mathbb{E}\left[ \|F_j - \Phi_j\|_{L^2(\mu)}^2 + \sum_{i=1}^{N+d} \|\partial_i F_j - \partial_i \Phi_j\|_{L^2(\mu)}^2 \right] = \mathbb{E}\left[ \|F_j - \Phi_j\|_{L^2(\mu)}^2 \right] + \sum_{i=1}^{N+d} \mathbb{E}\left[ \|\partial_i F_j - \partial_i \Phi_j\|_{L^2(\mu)}^2 \right] \tag{24}$$

by estimating the summands separately. To achieve this, we first compute $\mathbb{E}[\Phi_j(\boldsymbol{x}, \boldsymbol{z})]$ and $\mathbb{E}[\partial_i \Phi_j(\boldsymbol{x}, \boldsymbol{z})]$. Indeed, inserting the definitions, using independence and representation (19) yields

$$\mathbb{E}[\Phi_j(\boldsymbol{x}, \boldsymbol{z})] = \mathbb{E}[W_1 \cos(B_1 + \mathbf{A}_1 \cdot (\boldsymbol{x}, \boldsymbol{z}))]$$

$$= \|\widehat{F_j}\|_1 \mathbb{E}\left[\left(\frac{\operatorname{Re}[\widehat{F_j}](\mathbf{U}_1^{\operatorname{Re}})}{|\operatorname{Re}[\widehat{F_j}](\mathbf{U}_1^{\operatorname{Re}})|} Z_1 + \frac{\operatorname{Im}[\widehat{F_j}](\mathbf{U}_1^{\operatorname{Im}})}{|\operatorname{Im}[\widehat{F_j}](\mathbf{U}_1^{\operatorname{Im}})|}(1 - Z_1)\right)\right.$$

$$\left. \cos\left(\frac{\pi}{2}(1 - Z_1) + 2\pi(Z_1 \mathbf{U}_1^{\operatorname{Re}} + (1 - Z_1)\mathbf{U}_i^{\operatorname{Im}}) \cdot (\boldsymbol{x}, \boldsymbol{z})\right)\right]$$

$$= \|\widehat{F_j}\|_1 \left(\mathbb{P}(Z_1 = 1)\mathbb{E}\left[\frac{\operatorname{Re}[\widehat{F_j}](\mathbf{U}_1^{\operatorname{Re}})}{|\operatorname{Re}[\widehat{F_j}](\mathbf{U}_1^{\operatorname{Re}})|} \cos(2\pi \mathbf{U}_1^{\operatorname{Re}} \cdot (\boldsymbol{x}, \boldsymbol{z}))\right]\right.$$

$$\left. + \mathbb{P}(Z_1 = 0)\mathbb{E}\left[\frac{\operatorname{Im}[\widehat{F_j}](\mathbf{U}_1^{\operatorname{Im}})}{|\operatorname{Im}[\widehat{F_j}](\mathbf{U}_1^{\operatorname{Im}})|} \cos\left(\frac{\pi}{2} + 2\pi \mathbf{U}_1^{\operatorname{Im}} \cdot (\boldsymbol{x}, \boldsymbol{z})\right)\right]\right)$$

$$= \int_{\mathbb{R}^N \times \mathbb{R}^d} \operatorname{Re}[\widehat{F_j}](\boldsymbol{\xi}) \cos(2\pi \boldsymbol{\xi} \cdot (\boldsymbol{x}, \boldsymbol{z})) \mathrm{d}\boldsymbol{\xi} + \int_{\mathbb{R}^N \times \mathbb{R}^d} \operatorname{Im}[\widehat{F_j}](\boldsymbol{\xi}) \cos(\frac{\pi}{2} + 2\pi \boldsymbol{\xi} \cdot (\boldsymbol{x}, \boldsymbol{z})) \mathrm{d}\boldsymbol{\xi}$$

$$= F_j(\boldsymbol{x}, \boldsymbol{z}).$$

Analogously, using the representation (20) for the partial derivative $\partial_i F_j$ instead, we obtain

$$\mathbb{E}[\partial_i \Phi_j(\boldsymbol{x}, \boldsymbol{z})] = -\mathbb{E}[W_1 A_{1,i} \sin(B_1 + \mathbf{A}_1 \cdot (\boldsymbol{x}, \boldsymbol{z}))]$$

$$= -2\pi \|\widehat{F_j}\|_1 \left(\mathbb{P}(Z_1 = 1)\mathbb{E}\left[\frac{\operatorname{Re}[\widehat{F_j}](\mathbf{U}_1^{\operatorname{Re}})}{|\operatorname{Re}[\widehat{F_j}](\mathbf{U}_1^{\operatorname{Re}})|} U_{1,i}^{\operatorname{Re}} \sin(2\pi \mathbf{U}_1^{\operatorname{Re}} \cdot (\boldsymbol{x}, \boldsymbol{z}))\right]\right.$$

$$\left. + \mathbb{P}(Z_1 = 0)\mathbb{E}\left[\frac{\operatorname{Im}[\widehat{F_j}](\mathbf{U}_1^{\operatorname{Im}})}{|\operatorname{Im}[\widehat{F_j}](\mathbf{U}_1^{\operatorname{Im}})|} U_{1,i}^{\operatorname{Im}} \sin\left(\frac{\pi}{2} + 2\pi \mathbf{U}_1^{\operatorname{Im}} \cdot (\boldsymbol{x}, \boldsymbol{z})\right)\right]\right)$$

$$= -2\pi \left(\int_{\mathbb{R}^N \times \mathbb{R}^d} \xi_i \operatorname{Re}[\widehat{F_j}](\boldsymbol{\xi}) \sin(2\pi \boldsymbol{\xi} \cdot (\boldsymbol{x}, \boldsymbol{z})) \mathrm{d}\boldsymbol{\xi} + \int_{\mathbb{R}^N \times \mathbb{R}^d} \xi_i \operatorname{Im}[\widehat{F_j}](\boldsymbol{\xi}) \sin(\frac{\pi}{2} + 2\pi \boldsymbol{\xi} \cdot (\boldsymbol{x}, \boldsymbol{z})) \mathrm{d}\boldsymbol{\xi}\right)$$

$$= \partial_i F_j(\boldsymbol{x}, \boldsymbol{z}).$$

$$(25)$$

Therefore, we may estimate the first expectation in (24) as follows:

$$\mathbb{E}\left[\|F_j - \Phi_j\|_{L^2(\mu)}^2\right] = \mathbb{E}\left[\int_{\mathbb{R}^N \times \mathbb{R}^d} |F_j(\boldsymbol{x}, \boldsymbol{z}) - \Phi_j(\boldsymbol{x}, \boldsymbol{z})|^2 \mu(\mathrm{d}\boldsymbol{x}, \mathrm{d}\boldsymbol{z})\right] = \int_{\mathbb{R}^N \times \mathbb{R}^d} \mathbb{V}[\Phi_j(\boldsymbol{x}, \boldsymbol{z})] \mu(\mathrm{d}\boldsymbol{x}, \mathrm{d}\boldsymbol{z})$$

$$= \frac{1}{n^2} \int_{\mathbb{R}^N \times \mathbb{R}^d} \mathbb{V}\left[\sum_{i=1}^n W_i \cos(B_i + \mathbf{A}_i \cdot (\boldsymbol{x}, \boldsymbol{z}))\right] \mu(\mathrm{d}\boldsymbol{x}, \mathrm{d}\boldsymbol{z})$$

$$= \frac{1}{n} \int_{\mathbb{R}^N \times \mathbb{R}^d} \mathbb{V}\left[W_1 \cos(B_1 + \mathbf{A}_1 \cdot (\boldsymbol{x}, \boldsymbol{z}))\right] \mu(\mathrm{d}\boldsymbol{x}, \mathrm{d}\boldsymbol{z})$$

$$\leq \frac{1}{n} \int_{\mathbb{R}^N \times \mathbb{R}^d} \mathbb{E}\left[(W_1 \cos(B_1 + \mathbf{A}_1 \cdot (\boldsymbol{x}, \boldsymbol{z})))^2\right] \mu(\mathrm{d}\boldsymbol{x}, \mathrm{d}\boldsymbol{z})$$

$$\leq \frac{1}{n} \mathbb{E}\left[W_1^2\right] = \frac{1}{n} \|\widehat{F_j}\|_1^2.$$

$$(26)$$

For the partial derivatives, we obtain analogously

$$
\begin{aligned}
\mathbb{E}\left[\|\partial_i F_j - \partial_i \Phi_j\|_{L^2(\mu)}^2\right] &= \int_{\mathbb{R}^N \times \mathbb{R}^d} \mathbb{V}[\partial_i \Phi_j(\boldsymbol{x}, \boldsymbol{z})]\mu(\mathrm{d}\boldsymbol{x}, \mathrm{d}\boldsymbol{z}) \\
&= \frac{1}{n^2} \int_{\mathbb{R}^N \times \mathbb{R}^d} \mathbb{V}\left[\sum_{k=1}^n W_k A_{k,i} \sin(B_k + \mathbf{A}_k \cdot (\boldsymbol{x}, \boldsymbol{z}))\right] \mu(\mathrm{d}\boldsymbol{x}, \mathrm{d}\boldsymbol{z}) \\
&= \frac{1}{n} \int_{\mathbb{R}^N \times \mathbb{R}^d} \mathbb{V}\left[W_1 A_{1,i} \sin(B_1 + \mathbf{A}_1 \cdot (\boldsymbol{x}, \boldsymbol{z}))\right] \mu(\mathrm{d}\boldsymbol{x}, \mathrm{d}\boldsymbol{z}) \\
&\leq \frac{1}{n} \int_{\mathbb{R}^N \times \mathbb{R}^d} \mathbb{E}\left[(W_1 A_{1,i} \sin(B_1 + \mathbf{A}_1 \cdot (\boldsymbol{x}, \boldsymbol{z})))^2\right] \mu(\mathrm{d}\boldsymbol{x}, \mathrm{d}\boldsymbol{z}) \\
&\leq \frac{1}{n} \mathbb{E}\left[W_1^2 A_{1,i}^2\right] = \frac{1}{n}\|\widehat{F_j}\|_1^2 \mathbb{E}\left[A_{1,i}^2\right] = \frac{4\pi^2}{n}\|\widehat{F_j}\|_1 \int_{\mathbb{R}^N \times \mathbb{R}^d} \xi_i^2 |\widehat{F_j}(\boldsymbol{\xi})|\mathrm{d}\boldsymbol{\xi},
\end{aligned}
$$

(27)

where we used that $\mathbb{E}\left[A_{1,i}^2\right] = 4\pi^2 \|\widehat{F_j}\|_1^{-1} \int_{\mathbb{R}^N \times \mathbb{R}^d} \xi_i^2 |\widehat{F_j}(\boldsymbol{\xi})|\mathrm{d}\boldsymbol{\xi}$.

In particular, (26) and (27) imply that there exists a scenario $\omega \in \Omega$ such that $\Phi_j^\omega(\boldsymbol{x}, \boldsymbol{z}) = \frac{1}{n}\sum_{i=1}^n W_i(\omega)\cos(B_i(\omega) + \mathbf{A}_i(\omega) \cdot (\boldsymbol{x}, \boldsymbol{z}))$ satisfies

$$
\|F_j - \Phi_j^\omega\|_{L^2(\mu)}^2 + \sum_{i=1}^{N+d} \|\partial_i F_j - \partial_i \Phi_j^\omega\|_{L^2(\mu)}^2 \leq \frac{C_j}{n},
$$

(28)

with $C_j = \|\widehat{F_j}\|_1^2 + 4\pi^2 \|\widehat{F_j}\|_1 \int_{\mathbb{R}^N \times \mathbb{R}^d} \sum_{i=1}^{N+d} \xi_i^2 |\widehat{F_j}(\boldsymbol{\xi})|\mathrm{d}\boldsymbol{\xi}$. Finally, $\boldsymbol{\theta} = (\boldsymbol{\theta}^1, \ldots, \boldsymbol{\theta}^N)$ can then be constructed by setting $\boldsymbol{\theta}^j = (\mathbf{A}_i(\omega), B_i(\omega), \arccos(\frac{W_i(\omega)}{R}))_{i=1,\ldots,n}$, which guarantees that $\Phi_j^\omega = \bar{F}_{R,j}^{n,\boldsymbol{\theta}}$ and so the proposition follows. $\qquad\square$

## B.3 PROOF OF COROLLARY 4.3

The proof of this corollary requires the following lemma, which extends Gonon (2024, Lemma 4.10).

**Lemma B.1.** *Let $d, n, q \in \mathbb{N}$, let $M_1, M_2 > 0$, let $U$ be a non-negative random variable, and let $Y_1, \ldots, Y_n$ be i.i.d. $\mathbb{R}^d$-valued random variables. Suppose $\mathbb{E}[U] \leq M_1$ and $\mathbb{E}[|Y_1|^q] \leq M_2$. Then*

$$
\mathbb{P}\left[U \leq 3M_1, \max_{i=1,\ldots,n}|Y_i| \leq (3nM_2)^{\frac{1}{q}}\right] > 0.
$$

*Proof.* The proof mimics that of in Gonon (2024, Lemma 4.10) by replacing the use of Markov's inequality for $q = 1$ by the more general version:

$$
\mathbb{P}[|Y_1| > (3nM_2)^{\frac{1}{q}}] \leq \frac{\mathbb{E}[|Y_1|^q]}{3nM_2} \leq \frac{1}{3n}.
$$

$\qquad\square$

*Proof of the corollary.* The corollary follows by replacing the argument leading to (28) in the proof of Proposition 4.2 by Lemma B.1 and by noticing that

$$
\mathbb{E}\left[\|\mathbf{A}_1\|^q\right] = (2\pi)^q \|\widehat{F_j}\|_1^{-1} \int_{\mathbb{R}^N \times \mathbb{R}^d} \|\boldsymbol{\xi}\|^q |\widehat{F_j}(\boldsymbol{\xi})|\mathrm{d}\boldsymbol{\xi}.
$$

$\qquad\square$

## B.4 PROOF OF PROPOSITION 4.4

*Proof.* It follows by combining the proof of Proposition 4.2 with the proof of Theorem 3 in Gonon & Jacquier (2025). More specifically, the same proof can be used as for Proposition 4.2, except that

we need to replace the $L^2(\mu)$ error bounds in (26) and (27) by uniform bounds. For (26), we can follow precisely the proof of Theorem 3 in Gonon & Jacquier (2025) to obtain

$$\left\|\bar{F}_{R,j}^{n,\boldsymbol{\theta}} - F_j\right\|_{\infty,M} \leq \frac{C_j^{\infty,0}}{\sqrt{n}} \tag{29}$$

with $C_j^{\infty,0} = 2(\pi+1)\|\widehat{F_j}\|_1 + 8\pi M(N+d)^{\frac{1}{2}}\|\widehat{F_j}\|_1^{\frac{1}{2}}\left(\int_{\mathbb{R}^N\times\mathbb{R}^d}\sum_{i=1}^{N+d}\xi_i^2|\widehat{F_j}(\boldsymbol{\xi})|\mathrm{d}\boldsymbol{\xi}\right)^{1/2}$. Next, we

turn to the derivatives, that is, we aim to estimate $\left\|\partial_k\bar{F}_{R,j}^{n,\boldsymbol{\theta}} - \partial_k F_j\right\|_{\infty,M}$. Also in this case, we may proceed as in the proof of Theorem 3 in Gonon & Jacquier (2025) and apply the same estimates to the random variables $U_{i,(\boldsymbol{x},\boldsymbol{z})} = W_i A_{i,k}\sin(B_i + \mathbf{A}_i \cdot (\boldsymbol{x},\boldsymbol{z}))$. Let $\varepsilon_1,\ldots,\varepsilon_n$ be i.i.d. Rademacher random variables independent of $\mathbf{A} = (\mathbf{A}_1,\ldots,\mathbf{A}_n)$ and $\mathbf{B} = (B_1,\ldots,B_n)$. Symmetrisation and independence then yield

$$\left\|\partial_i\bar{F}_{R,j}^{n,\boldsymbol{\theta}} - \partial_i F_j\right\|_{\infty,M} = \mathbb{E}\left[\sup_{(\boldsymbol{x},\boldsymbol{z})\in[-M,M]^{N+d}}\left|\frac{1}{n}\sum_{i=1}^{n}\left(U_{i,(\boldsymbol{x},\boldsymbol{z})} - \mathbb{E}[U_{i,(\boldsymbol{x},\boldsymbol{z})}]\right)\right|\right]$$

$$\leq 2\mathbb{E}\left[\sup_{(\boldsymbol{x},\boldsymbol{z})\in[-M,M]^{N+d}}\left|\frac{1}{n}\sum_{i=1}^{n}\varepsilon_i U_{i,(\boldsymbol{x},\boldsymbol{z})}\right|\right]$$

$$= 2\mathbb{E}\left[\mathbb{E}\left[\sup_{(\boldsymbol{x},\boldsymbol{z})\in[-M,M]^{N+d}}\left|\frac{1}{n}\sum_{i=1}^{n}\varepsilon_i w_i a_{i,k}\sin(b_i + \mathbf{a}_i \cdot (\boldsymbol{x},\boldsymbol{z}))\right|\right]_{(\mathbf{w},\mathbf{a},\mathbf{b})=(\mathbf{W},\mathbf{A},\mathbf{B})}\right].$$

Now fix $\boldsymbol{a} = (\boldsymbol{a}_1,\ldots,\boldsymbol{a}_n) \in (\mathbb{R}^N\times\mathbb{R}^d)^n$, $\mathbf{b} = (b_1,\ldots,b_n) \in \mathbb{R}^n$, $\mathbf{w} = (w_1,\ldots,w_n) \in \mathbb{R}^n$ and denote

$$\mathcal{T} := \{(w_i a_{i,k}(b_i + \mathbf{a}_i \cdot (\boldsymbol{x},\boldsymbol{z})))_{i=1,\ldots,n} : (\boldsymbol{x},\boldsymbol{z}) \in [-M,M]^{N+d}\},$$

$$\varrho_i(x) := w_i a_{i,k}\sin(\frac{x}{w_i a_{i,k}}), \quad x \in \mathbb{R},$$

for $i = 1,\ldots,n$. Then, using the definitions in the first step, the comparison theorem (Ledoux & Talagrand, 2013, Theorem 4.12) in the second step (note $\varrho_i(0) = 0$ and $\varrho_i$ is 1-Lipschitz), and standard Rademacher estimates (see, e.g., Gonon (2023)), we obtain

$$\mathbb{E}\left[\sup_{(\boldsymbol{x},\boldsymbol{z})\in[-M,M]^{N+d}}\left|\frac{1}{n}\sum_{i=1}^{n}\varepsilon_i w_i a_{i,k}\sin(b_i + \mathbf{a}_i \cdot (\boldsymbol{x},\boldsymbol{z}))\right|\right]$$

$$= \mathbb{E}\left[\sup_{\mathbf{t}\in\mathcal{T}}\left|\frac{1}{n}\sum_{i=1}^{n}\varepsilon_i\varrho_i(t_i)\right|\right] \leq 2\mathbb{E}\left[\sup_{\mathbf{t}\in\mathcal{T}}\left|\frac{1}{n}\sum_{i=1}^{n}\varepsilon_i t_i\right|\right]$$

$$= 2\mathbb{E}\left[\sup_{(\boldsymbol{x},\boldsymbol{z})\in[-M,M]^{N+d}}\left|\frac{1}{n}\sum_{i=1}^{n}\varepsilon_i(w_i a_{i,k}(b_i + \mathbf{a}_i \cdot (\boldsymbol{x},\boldsymbol{z}))\right|\right]$$

$$\leq 2\mathbb{E}\left[\left|\frac{1}{n}\sum_{i=1}^{n}\varepsilon_i w_i a_{i,k}b_i\right|\right] + 2\mathbb{E}\left[\sup_{(\boldsymbol{x},\boldsymbol{z})\in[-M,M]^{N+d}}\left|(\boldsymbol{x},\boldsymbol{z})\cdot\frac{1}{n}\sum_{i=1}^{n}\varepsilon_i w_i a_{i,k}\mathbf{a}_i\right|\right]$$

$$\leq \frac{2}{n}\left(\sum_{i=1}^{n}w_i^2 a_{i,k}^2 b_i^2\right)^{1/2} + \frac{2M}{n}\sum_{l=1}^{N+d}\left(\sum_{i=1}^{n}w_i^2 a_{i,k}^2 a_{i,l}^2\right)^{1/2}.$$

Putting everything together, we obtain

$$\left\|\partial_i\bar{F}_{R,j}^{n,\boldsymbol{\theta}} - \partial_i F_j\right\|_{\infty,M} \leq 2\mathbb{E}\left[\frac{2}{n}\left(\sum_{i=1}^{n}W_i^2 A_{i,k}^2 B_i^2\right)^{1/2} + \frac{2M}{n}\sum_{l=1}^{N+d}\left(\sum_{i=1}^{n}W_i^2 A_{i,k}^2 A_{i,l}^2\right)^{1/2}\right]$$

$$\leq \frac{4}{\sqrt{n}}\left(\mathbb{E}\left[W_i^2 A_{i,k}^2 B_i^2\right]^{1/2} + M(N+d)^{1/2}\left(\sum_{l=1}^{N+d}\mathbb{E}\left[W_i^2 A_{i,k}^2 A_{i,l}^2\right]\right)^{1/2}\right)$$

$$\leq \frac{C_j^{\infty,k}}{\sqrt{n}},$$

with $C_j^{\infty,k} = 4\pi^2 \|\widehat{F_j}\|_1^{1/2} \left( \left( \int_{\mathbb{R}^N \times \mathbb{R}^d} \xi_k^2 |\widehat{F_j}(\boldsymbol{\xi})| \mathrm{d}\boldsymbol{\xi} \right)^{1/2} + 4M(N+d)^{1/2} \left( \int_{\mathbb{R}^N \times \mathbb{R}^d} \xi_k^2 \|\boldsymbol{\xi}\|^2 |\widehat{F_j}(\boldsymbol{\xi})| \mathrm{d}\boldsymbol{\xi} \right)^{1/2} \right)$.

Here, the last estimate follows from the inequality

$$\mathbb{E}\left[ W_i^2 A_{i,k}^2 B_i^2 \right] \leq \pi^4 \|\widehat{F_j}\|_1 \int_{\mathbb{R}^N \times \mathbb{R}^d} \xi_k^2 |\widehat{F_j}(\boldsymbol{\xi})| \mathrm{d}\boldsymbol{\xi}$$

and

$$\mathbb{E}\left[ W_i^2 A_{i,k}^2 A_{i,l}^2 \right] = 16\pi^4 \|\widehat{F_j}\|_1 \int_{\mathbb{R}^N \times \mathbb{R}^d} \xi_k^2 \xi_l^2 |\widehat{F_j}(\boldsymbol{\xi})| \mathrm{d}\boldsymbol{\xi}.$$

Overall, we obtain (9) with $C_j^\infty \geq \sum_{k=0}^{N+d} C_j^{\infty,k}$ chosen as

$$C_j^\infty = 2(\pi+1)\|\widehat{F_j}\|_1 + (8\pi M + 4\pi^2)(N+d)^{\frac{1}{2}} \|\widehat{F_j}\|_1^{\frac{1}{2}} I_{2,j}^{1/2} + 16M\pi^2(N+d)\|\widehat{F_j}\|_1^{1/2} I_{4,j}^{1/2}.$$

$\square$

### B.5 PROOF OF COROLLARY 4.5

*Proof.* First, extending the proof of Corollary 4 in Gonon & Jacquier (2025), we show that $F_j$ can be approximated on $\mathcal{X}$ up to error $\frac{\varepsilon}{2}$ in $C^1$-norm by a function in $C_c^\infty(\mathbb{R}^N \times \mathbb{R}^d)$. Indeed, first let $M > 0$ be such that $\mathcal{X} \subset [-M, M]^{N+d}$. Then, classical approximation results (see, e.g., Whitney, 1934, Lemma 5) imply that there exists a smooth function $h \colon \mathbb{R}^N \times \mathbb{R}^d \to \mathbb{R}$ such that

$$\sup_{(\boldsymbol{x}, \boldsymbol{z}) \in \mathcal{X}} |F_j(\boldsymbol{x}, \boldsymbol{z}) - h(\boldsymbol{x}, \boldsymbol{z})| + \|\nabla F_j(\boldsymbol{x}, \boldsymbol{z}) - \nabla h(\boldsymbol{x}, \boldsymbol{z})\| \leq \frac{\varepsilon}{2}. \tag{30}$$

Without loss of generality we may assume that $h \in C_c^\infty(\mathbb{R}^N \times \mathbb{R}^d)$. Otherwise, we multiply $h$ with a cutoff function $\psi \in C_c^\infty(\mathbb{R}^N \times \mathbb{R}^d)$ which is equal to 1 in an open set $U$ with $\mathcal{X} \subset U$ (see, e.g., Hörmander, 1990, Theorem 1.4.1); thereby preserving (30).

In the next step, we now apply Proposition 4.4 to $h$. Since $h$ is a Schwartz function, its Fourier transform $\widehat{h}$ is also a Schwartz function and thus $h$ is integrable and

$$\int_{\mathbb{R}^N \times \mathbb{R}^d} (1 + \|\boldsymbol{\xi}\|^4) |\widehat{h}(\boldsymbol{\xi})| \mathrm{d}\boldsymbol{\xi} < \infty.$$

In particular, $h \in \mathcal{F}_R$ for $R > 0$ large enough and, as $h$ is a Schwartz function, also $\partial_i h \in \mathcal{F}$ for all $i$. Thus, the hypotheses of Proposition 4.4 are satisfied and we obtain that there exist $n \in \mathbb{N}$ and $\boldsymbol{\theta} \in \Theta$ such that

$$\left\| \bar{F}_{R,j}^{n,\boldsymbol{\theta}} - h \right\|_{\infty,M} + \sum_{i=1}^{N+d} \left\| \partial_i \bar{F}_{R,j}^{n,\boldsymbol{\theta}} - \partial_i h \right\|_{\infty,M} \leq \frac{\varepsilon}{2}.$$

This estimate together with (30) then imply

$$\sup_{(\boldsymbol{x}, \boldsymbol{z}) \in \mathcal{X}} |F_j(\boldsymbol{x}, \boldsymbol{z}) - \bar{F}_{R,j}^{n,\boldsymbol{\theta}}(\boldsymbol{x}, \boldsymbol{z})| + \|\nabla F_j(\boldsymbol{x}, \boldsymbol{z}) - \nabla \bar{F}_{R,j}^{n,\boldsymbol{\theta}}(\boldsymbol{x}, \boldsymbol{z})\|$$

$$\leq \sup_{(\boldsymbol{x}, \boldsymbol{z}) \in \mathcal{X}} |F_j(\boldsymbol{x}, \boldsymbol{z}) - h(\boldsymbol{x}, \boldsymbol{z})| + \|\nabla F_j(\boldsymbol{x}, \boldsymbol{z}) - \nabla h(\boldsymbol{x}, \boldsymbol{z})\|$$

$$+ \sup_{(\boldsymbol{x}, \boldsymbol{z}) \in \mathcal{X}} |h(\boldsymbol{x}, \boldsymbol{z}) - \bar{F}_{R,j}^{n,\boldsymbol{\theta}}(\boldsymbol{x}, \boldsymbol{z})| + \|\nabla \bar{F}_{R,j}^{n,\boldsymbol{\theta}}(\boldsymbol{x}, \boldsymbol{z}) - \nabla h(\boldsymbol{x}, \boldsymbol{z})\|$$

$$\leq \sup_{(\boldsymbol{x}, \boldsymbol{z}) \in \mathcal{X}} |F_j(\boldsymbol{x}, \boldsymbol{z}) - h(\boldsymbol{x}, \boldsymbol{z})| + \|\nabla F_j(\boldsymbol{x}, \boldsymbol{z}) - \nabla h(\boldsymbol{x}, \boldsymbol{z})\|$$

$$+ \sup_{(\boldsymbol{x}, \boldsymbol{z}) \in \mathcal{X}} |\bar{F}_{R,j}^{n,\boldsymbol{\theta}}(\boldsymbol{x}, \boldsymbol{z}) - h(\boldsymbol{x}, \boldsymbol{z})| + \sum_{i=1}^{N+d} |\partial_i \bar{F}_{R,j}^{n,\boldsymbol{\theta}}(\boldsymbol{x}, \boldsymbol{z}) - \partial_i h(\boldsymbol{x}, \boldsymbol{z})|$$

$$\leq \sup_{(\boldsymbol{x}, \boldsymbol{z}) \in \mathcal{X}} |F_j(\boldsymbol{x}, \boldsymbol{z}) - h(\boldsymbol{x}, \boldsymbol{z})| + \|\nabla F_j(\boldsymbol{x}, \boldsymbol{z}) - \nabla h(\boldsymbol{x}, \boldsymbol{z})\|$$

$$+ \left\| \bar{F}_{R,j}^{n,\boldsymbol{\theta}} - h \right\|_{\infty,M} + \sum_{i=1}^{N+d} \left\| \partial_i \bar{F}_{R,j}^{n,\boldsymbol{\theta}} - \partial_i h \right\|_{\infty,M} \leq \varepsilon,$$

where we used that

$$\|\nabla \bar{F}_{R,j}^{n,\boldsymbol{\theta}}(\boldsymbol{x},\boldsymbol{z})-\nabla h(\boldsymbol{x},\boldsymbol{z})\| = \left(\sum_{i=1}^{N+d} |\partial_i \bar{F}_{R,j}^{n,\boldsymbol{\theta}}(\boldsymbol{x},\boldsymbol{z}) - \partial_i h(\boldsymbol{x},\boldsymbol{z})|^2\right)^{1/2} \leq \sum_{i=1}^{N+d} |\partial_i \bar{F}_{R,j}^{n,\boldsymbol{\theta}}(\boldsymbol{x},\boldsymbol{z})-\partial_i h(\boldsymbol{x},\boldsymbol{z})|,$$

since $\|\boldsymbol{y}\|_2 \leq \|\boldsymbol{y}\|_1$ for all $\boldsymbol{y} \in \mathbb{R}^{N+d}$. $\qquad\square$

## C  PROOFS FOR SECTION 4.2

### C.1  PROOF OF THEOREM 4.6

*Proof.* Choose $M$ such that $B_N \times D_d \subset [-M, M]^{N+d}$ and $[-R, R]^N \times D_d \subset [-M, M]^{N+d}$. Firstly, our hypotheses on $F$ guarantee that $F$ satisfies the hypotheses of Proposition 4.4. Hence, there exists $\boldsymbol{\theta} \in \boldsymbol{\Theta}$ such that for any $j \in \{1, \ldots, N\}$,

$$\left\|\bar{F}_{R,j}^{n,\boldsymbol{\theta}} - F_j\right\|_{\infty,M} + \sum_{i=1}^{N+d} \left\|\partial_i \bar{F}_{R,j}^{n,\boldsymbol{\theta}} - \partial_i F_j\right\|_{\infty,M} \leq \frac{C_j^\infty}{\sqrt{n}}. \tag{31}$$

Then, for all $\boldsymbol{x} \in [-M, M]^N, \boldsymbol{z} \in D_d$

$$\|\nabla_{\boldsymbol{x}} \bar{F}_R^{n,\boldsymbol{\theta}}(\boldsymbol{x},\boldsymbol{z})\|_2 \leq \|\nabla_{\boldsymbol{x}} \bar{F}_R^{n,\boldsymbol{\theta}}(\boldsymbol{x},\boldsymbol{z}) - \nabla_{\boldsymbol{x}} F(\boldsymbol{x},\boldsymbol{z})\|_2 + \|\nabla_{\boldsymbol{x}} F(\boldsymbol{x},\boldsymbol{z})\|_2$$

$$\leq \left(\sum_{i,j=1}^{N} |\partial_i \bar{F}_{R,j}^{n,\boldsymbol{\theta}}(\boldsymbol{x},\boldsymbol{z}) - \partial_i F_j(\boldsymbol{x},\boldsymbol{z})|^2\right)^{1/2} + \lambda \tag{32}$$

$$\leq N \frac{\max_{j=1,\ldots,N} C_j^\infty}{\sqrt{n}} + \lambda.$$

Therefore, using that $\max_{\boldsymbol{x}\in[-M,M]^N} \|\nabla_{\boldsymbol{x}} \bar{F}_R^{n,\boldsymbol{\theta}}(\boldsymbol{x},\boldsymbol{z})\|_2$ is the best Lipschitz-constant for $\bar{F}_R^{n,\boldsymbol{\theta}}$ on $[-M, M]^N$ for any given $\boldsymbol{z} \in D_d$, we obtain for all $\boldsymbol{x} \in [-M, M]^N, \boldsymbol{z} \in D_d$ that

$$\|\bar{F}_R^{n,\boldsymbol{\theta}}(\boldsymbol{x}^1,\boldsymbol{z}) - \bar{F}_R^{n,\boldsymbol{\theta}}(\boldsymbol{x}^2,\boldsymbol{z})\|^2 \leq \|\boldsymbol{x}^1 - \boldsymbol{x}^2\|^2 \max_{\boldsymbol{x}\in[-M,M]^N} \|\nabla_{\boldsymbol{x}} \bar{F}_R^{n,\boldsymbol{\theta}}(\boldsymbol{x},\boldsymbol{z})\|_2^2$$

$$\leq \left(N \frac{\max_{j=1,\ldots,N} C_j^\infty}{\sqrt{n}} + \lambda\right)^2 \|\boldsymbol{x}^1 - \boldsymbol{x}^2\|^2.$$

In particular, for $n$ satisfying $N^2 \frac{(\max_{j=1,\ldots,N} C_j^\infty)^2}{(1-\lambda)^2} < n$ we obtain that $\bar{F}_R^{n,\boldsymbol{\theta}} \colon B_R \times D_d \to B_R$, with $B_R = \{\boldsymbol{x} \in \mathbb{R}^N \colon \|\boldsymbol{x}\| \leq R\sqrt{N}\}$, is contractive in the first argument, hence the system (4) has the echo state property by Gonon et al. (2020, Proposition 1).

By the relation between the Lipschitz-constant and the maximal norm of the Jacobian, the assumption $\|\nabla_{\boldsymbol{x}} F(\boldsymbol{x},\boldsymbol{z})\|_2 \leq \lambda$ guarantees that $F(\cdot, \boldsymbol{z})$ is $\lambda$-contractive for any $\boldsymbol{z} \in D_d$. Hence, we may estimate

$$\|U^F(\boldsymbol{z})_t - \bar{U}(\boldsymbol{z})_t\| = \|\boldsymbol{x}_t - \hat{\boldsymbol{x}}_t\| = \left\|F(\boldsymbol{x}_{t-1}, \boldsymbol{z}_t) - \bar{F}_R^{n,\boldsymbol{\theta}}(\hat{\boldsymbol{x}}_{t-1}, \boldsymbol{z}_t)\right\|$$

$$\leq \|F(\boldsymbol{x}_{t-1}, \boldsymbol{z}_t) - F(\hat{\boldsymbol{x}}_{t-1}, \boldsymbol{z}_t)\| + \left\|F(\hat{\boldsymbol{x}}_{t-1}, \boldsymbol{z}_t) - \bar{F}_R^{n,\boldsymbol{\theta}}(\hat{\boldsymbol{x}}_{t-1}, \boldsymbol{z}_t)\right\|$$

$$\leq \lambda \|\boldsymbol{x}_{t-1} - \hat{\boldsymbol{x}}_{t-1}\| + \left(\sum_{j=1}^{N} \left\|\bar{F}_{R,j}^{n,\boldsymbol{\theta}} - F_j\right\|_{\infty,M}^2\right)^{1/2} \tag{33}$$

$$\leq \lambda \|\boldsymbol{x}_{t-1} - \hat{\boldsymbol{x}}_{t-1}\| + \frac{\sqrt{N} \max_{j=1,\ldots,N} C_j^\infty}{\sqrt{n}}.$$

Iterating (33), we obtain

$$\|U^F(\boldsymbol{z})_t - \bar{U}(\boldsymbol{z})_t\| \leq \lambda^J \|\boldsymbol{x}_{t-J} - \hat{\boldsymbol{x}}_{t-J}\| + \sum_{k=1}^{J} \lambda^{k-1} \frac{\sqrt{N} \max_{j=1,\ldots,N} C_j^\infty}{\sqrt{n}}$$

$$\leq \lambda^J \sqrt{N}(M + R) + \sum_{k=0}^{J-1} \lambda^k \frac{\sqrt{N} \max_{j=1,\ldots,N} C_j^\infty}{\sqrt{n}}. \tag{34}$$

Letting $J \to \infty$, we thus arrive at the bound (12). $\qquad\square$

## C.2 PROOF OF LEMMA 4.7

The proof of Lemma 4.7 is related to the approach introduced in Gonon & Ortega (2020) and subsequently used, e.g., in Gonon et al. (2023); Gonon & Ortega (2021).

*Proof.* We start by constructing a partition of $\hat{x}_t$ as in the statement. If $N = 1$, we simply have $\hat{\boldsymbol{x}}_t = [\hat{x}_t] \in \mathbb{R}$. Next, we define the reservoir vector $\tilde{F}_{R,i:j} = (\tilde{F}_i, \ldots, \tilde{F}_j)$. Then, for $k = 1$, $j \in \{1, \ldots, l_1\}$, and $k = 2, \ldots, K-1$, $j \in \{l_{k-1}+1, \ldots, l_k\}$, we have $P_j \hat{\boldsymbol{x}}_t = [\hat{\boldsymbol{x}}_t^{(k+1)}, \ldots, \hat{\boldsymbol{x}}_t^{(K)}, 0, \ldots, 0]$ and $P_j \hat{\boldsymbol{x}}_t = 0$ for $j = l_{K-1} + 1, \ldots, N$. Inserting these choices into (15), we may rewrite the dynamics as

$$\hat{\boldsymbol{x}}_t^{(k)} = \tilde{F}_{l_{k-1}+1:l_k}([\hat{\boldsymbol{x}}_{t-1}^{(k+1)}, \ldots, \hat{\boldsymbol{x}}_{t-1}^{(K)}, 0, \ldots, 0], \boldsymbol{z}_t), \quad t \in \mathbb{Z}_-, \tag{35}$$

for $k = 1, \ldots, K-1$ and $\hat{\boldsymbol{x}}_t^{(K)} = \tilde{F}_{l_{K-1}+1:l_K}(0, \boldsymbol{z}_t)$. In particular, $\hat{\boldsymbol{x}}_t^{(K)} = \tilde{F}_{l_{K-1}+1:l_K}(0, \boldsymbol{z}_t)$, which depends only on $\boldsymbol{z}_t$, is explicitly given for all $t \in \mathbb{Z}_-$, and for all $k = 1, \ldots, K-1$, we see that $\hat{\boldsymbol{x}}_t^{(k)}$ only depends on $\hat{\boldsymbol{x}}_{t-1}^{(k+1)}, \ldots, \hat{\boldsymbol{x}}_{t-1}^{(K)}$. Thus, (15) admits a unique solution which can be explicitly obtained from the recursion (35), that is, for all $t \in \mathbb{Z}_-$, we have $\hat{\boldsymbol{x}}_t^{(K)} = \tilde{F}_{l_{K-1}+1:l_K}(0, \boldsymbol{z}_t)$, $\hat{\boldsymbol{x}}_t^{(K-1)} = \tilde{F}_{l_{K-2}+1:l_{K-1}}([\hat{\boldsymbol{x}}_{t-1}^{(K)}, 0, \ldots, 0], \boldsymbol{z}_t)$, $\ldots$, $\hat{\boldsymbol{x}}_t^{(1)} = \tilde{F}_{1:l_1}([\hat{\boldsymbol{x}}_{t-1}^{(2)}, \ldots, \hat{\boldsymbol{x}}_{t-1}^{(K)}, 0], \boldsymbol{z}_t)$. This proves that $\tilde{F}$ has the echo state property. $\qquad\square$

## C.3 PROOF OF THEOREM 4.8

*Proof.* Without loss of generality we may assume $\varepsilon \leq 1$, because proving (17) for $\varepsilon \leq 1$ also implies that (17) holds for $\varepsilon > 1$.

Let $H_U : (D_d)^{\mathbb{Z}_-} \to B_m$ be the functional associated to the filter $U$. Then, as in the proof of Gonon & Ortega (2021, Theorem 2.1), there exists $K \in \mathbb{N}$ and a continuous function $\bar{G} : (D_d)^{dK} \to B_m$ such that

$$\sup_{\boldsymbol{z} \in (D_d)^{\mathbb{Z}_-}} \left\| H_U(\boldsymbol{z}) - \bar{G}(\boldsymbol{z}_{-K+1}, \ldots, \boldsymbol{z}_0) \right\| < \frac{\varepsilon}{4}. \tag{36}$$

Moreover, e.g., by the argument in Gonon & Jacquier (2025, Corollary 4), there exists a function $G \in C_c^\infty((\mathbb{R}^d)^K, B_m)$ which satisfies

$$\sup_{\boldsymbol{z} \in (\mathbb{R}^d)^K} \left\| G(\boldsymbol{z}) - \bar{G}(\boldsymbol{z}) \right\| < \frac{\varepsilon}{4}. \tag{37}$$

Next, choose $N = (K-1)d + m$ and consider the recurrent QNN introduced in (4). Denote

$$\bar{F}_{R,j}^{n,\boldsymbol{\theta}}(\boldsymbol{x}, \boldsymbol{z}) = R - 2R[\mathbb{P}_1^{n,\boldsymbol{\theta}^j}(\boldsymbol{x}, \boldsymbol{z}) + \mathbb{P}_2^{n,\boldsymbol{\theta}^j}(\boldsymbol{x}, \boldsymbol{z})], \quad (\boldsymbol{x}, \boldsymbol{z}) \in \mathbb{R}^N \times \mathbb{R}^d \tag{38}$$

the update maps without preprocessing matrices. For $1 \leq i \leq j \leq N$, write $\bar{F}_{R,i:j}^{n,\boldsymbol{\theta}} = (\bar{F}_{R,i}^{n,\boldsymbol{\theta}}, \ldots, \bar{F}_{R,j}^{n,\boldsymbol{\theta}})$ and $l_k = m + (k-1)d$ for $k = 1, \ldots, K$. Define the constants

$$L_G = \max(\sqrt{d}, \sup_{\boldsymbol{z} \in (\mathbb{R}^d)^K} \|\nabla G(\boldsymbol{z})\|) + 1, \qquad C_G = 4L_G \left( \sum_{k=2}^{K} \sum_{j=1}^{K-k+1} (2L_G)^j \right)^{1/2}. \tag{39}$$

Then, as $G \in C_c^\infty((\mathbb{R}^d)^K)$ and the identity is smooth, Corollary 4.5 (applied componentwise) guarantees that there exist $n_K$, $R_K$ and $\boldsymbol{\theta}_K \in \Theta^d$ such that

$$\sup_{\boldsymbol{z} \in D_d} \|\bar{F}_{R_K, l_{K-1}+1:l_K}^{n_K, \boldsymbol{\theta}_K}(0, \boldsymbol{z}) - \boldsymbol{z}\| + \sup_{\boldsymbol{z} \in D_d} \|\nabla \bar{F}_{R_K, l_{K-1}+1:l_K}^{n_K, \boldsymbol{\theta}_K}(0, \boldsymbol{z}) - \mathbf{1}_d\| < \frac{\varepsilon}{C_G}, \tag{40}$$

and (recursively), for all $k = K-1, \ldots, 2$ there exist $n_k$, $R_k$ and $\boldsymbol{\theta}_k \in \Theta^d$ such that

$$\sup_{(\boldsymbol{x}, \boldsymbol{z}) \in [-R_{k+1}, R_{k+1}]^N \times D_d} \|\bar{F}_{R_k, l_{k-1}+1:l_k}^{n_k, \boldsymbol{\theta}_k}(\boldsymbol{x}, \boldsymbol{z}) - \boldsymbol{x}_{1:d}\| + \|\nabla \bar{F}_{R_k, l_{k-1}+1:l_k}^{n_k, \boldsymbol{\theta}_k}(\boldsymbol{x}, \boldsymbol{z}) - \mathbf{1}_d\| < \frac{\varepsilon}{C_G}, \tag{41}$$

and there exist $n_1$, $R_1$ and $\boldsymbol{\theta}_1 \in \boldsymbol{\Theta}^d$ such that

$$\sup_{([\boldsymbol{z}_{-K+1},\ldots,\boldsymbol{z}_{-1}],\boldsymbol{z}_0)\in[-R_2,R_2]^N\times D_d} \Big( \|\bar{F}_{R_1,1:m}^{n_1,\boldsymbol{\theta}_1}([\boldsymbol{z}_{-K+1},\ldots,\boldsymbol{z}_{-1},0],\boldsymbol{z}_0) - G(\boldsymbol{z}_{-K+1},\ldots,\boldsymbol{z}_0)\|$$

$$+ \|\nabla \bar{F}_{R_1,1:m}^{n_1,\boldsymbol{\theta}_1}([\boldsymbol{z}_{-K+1},\ldots,\boldsymbol{z}_{-1},0],\boldsymbol{z}_0) - \nabla G(\boldsymbol{z}_{-K+1},\ldots,\boldsymbol{z}_0)\| \Big) < \frac{\varepsilon}{4}. \tag{42}$$

Without loss of generality we may choose $R = R_1 = \ldots = R_K$, since we can always replace $R_k$ by $\max(R_k, R_{k+1})$ (and hence ultimately replace $R_1,\ldots,R_K$ by $R$) and absorb the change in an adjusted choice of parameters $\gamma^{i,j}$ (see representation (7)). Moreover, by a similar reasoning we may assume without loss of generality that $n = n_1 = \ldots = n_K$. Indeed, otherwise we may again choose $n$ to be the maximum of $n_1,\ldots,n_K$, replace $n_1,\ldots,n_K$ by $n$ and recover the same functions (7) by setting surplus terms $i > n_k$ to 0 by appropriate choice of $\gamma^{i,j}$. The extra factor $\frac{n}{n_k}$, in turn, can be absorbed by modifying the choice of $R$.

Denote by $L_k$ be the best Lipschitz constant for $\bar{F}_{R,l_{k-1}+1:l_k}^{n,\boldsymbol{\theta}_k}$. Then (40)–(42) imply that $L_k \leq \sqrt{d} + \varepsilon \leq L_G$ for $k = K,\ldots,2$ and $L_1 \leq \sup_{\boldsymbol{z}\in\mathbb{R}^d)^K} \|\nabla G(\boldsymbol{z})\| + 1 \leq L_G$. In particular, $L_G \geq \max(L_1,\ldots,L_K)$ is a bound on the Lipschitz constant for all QNNs $\bar{F}_{R,l_{k-1}+1:l_k}^{n,\boldsymbol{\theta}_k}$ and $G$. Partition $\hat{\boldsymbol{x}}_t = [\hat{\boldsymbol{x}}_t^{(1)},\ldots,\hat{\boldsymbol{x}}_t^{(K)}] \in \mathbb{R}^m \times (\mathbb{R}^d)^{K-1}$. Using the triangle inequality, we then obtain

$$\sup_{\boldsymbol{z}\in(D_d)^{K+1}} \left\| G(\boldsymbol{z}_{-K+1},\ldots,\boldsymbol{z}_0) - \hat{\boldsymbol{x}}_0^{(1)} \right\|$$

$$= \sup_{\boldsymbol{z}\in(D_d)^{(K+1)}} \left\| G(\boldsymbol{z}_{-K+1},\ldots,\boldsymbol{z}_0) - \bar{F}_{R,1:m}^{n,\boldsymbol{\theta}}([\hat{\boldsymbol{x}}_{-1}^{(2)},\ldots,\hat{\boldsymbol{x}}_{-1}^{(K)},0],\boldsymbol{z}_0) \right\|$$

$$\leq \left\| G(\boldsymbol{z}_{-K+1},\ldots,\boldsymbol{z}_0) - G([\hat{\boldsymbol{x}}_{-1}^{(2)},\ldots,\hat{\boldsymbol{x}}_{-1}^{(K)}],\boldsymbol{z}_0) \right\| \tag{43}$$

$$+ \left\| G([\hat{\boldsymbol{x}}_{-1}^{(2)},\ldots,\hat{\boldsymbol{x}}_{-1}^{(K)}],\boldsymbol{z}_0) - \bar{F}_{R,1:m}^{n,\boldsymbol{\theta}}([\hat{\boldsymbol{x}}_{-1}^{(2)},\ldots,\hat{\boldsymbol{x}}_{-1}^{(K)},0],\boldsymbol{z}_0) \right\|$$

$$\leq L_G \left\| (\boldsymbol{z}_{-K+1},\ldots,\boldsymbol{z}_0) - ([\hat{\boldsymbol{x}}_{-1}^{(2)},\ldots,\hat{\boldsymbol{x}}_{-1}^{(K)}],\boldsymbol{z}_0) \right\| + \frac{\varepsilon}{4}.$$

For the last norm, we write

$$\left\| (\boldsymbol{z}_{-K+1},\ldots,\boldsymbol{z}_0) - ([\hat{\boldsymbol{x}}_{-1}^{(2)},\ldots,\hat{\boldsymbol{x}}_{-1}^{(K)}],\boldsymbol{z}_0) \right\|^2 = \sum_{k=0}^{K-2} \left\| \boldsymbol{z}_{-k-1} - \hat{\boldsymbol{x}}_{-1}^{(K-k)} \right\|^2 = \sum_{k=2}^{K} \left\| \boldsymbol{z}_{-K+k-1} - \hat{\boldsymbol{x}}_{-1}^{(k)} \right\|^2.$$

We proceed by backward induction over $k$ to prove that for all $k = K,\ldots,2$ it holds

$$\left\| \boldsymbol{z}_{-K+k+t} - \hat{\boldsymbol{x}}_t^{(k)} \right\|^2 \leq \sum_{j=1}^{K-k+1} (2L_G)^j \frac{\varepsilon^2}{C_G^2},$$

for arbitrary $t \in \mathbb{Z}_-$. Indeed, we have

$$\left\| \boldsymbol{z}_{-K+k+t} - \hat{\boldsymbol{x}}_t^{(k)} \right\|^2 = \left\| \boldsymbol{z}_{-K+k+t} - \bar{F}_{R,l_{k-1}+1:l_k}^{n,\boldsymbol{\theta}}([\hat{\boldsymbol{x}}_{t-1}^{(k+1)},\ldots,\hat{\boldsymbol{x}}_{t-1}^{(K)},0,\ldots,0],\boldsymbol{z}_t) \right\|^2$$

and so for $k = K$ it follows that

$$\left\| \boldsymbol{z}_{-K+k+t} - \hat{\boldsymbol{x}}_t^{(k)} \right\|^2 = \left\| \boldsymbol{z}_t - \bar{F}_{R,l_{K-1}+1:l_K}^{n,\boldsymbol{\theta}}(0,\boldsymbol{z}_t) \right\|^2 \leq \frac{\varepsilon^2}{C_G^2} \leq 2L_G \frac{\varepsilon^2}{C_G^2}$$

Assume that the bound holds for a fixed $k \in \{K,\ldots,3\}$, then for $k-1$ we estimate (with the notation $f_{k-1} = \bar{F}_{R,l_{k-2}+1:l_{k-1}}^{n,\boldsymbol{\theta}}$)

$$\left\| \boldsymbol{z}_{-K+(k-1)+t} - \hat{\boldsymbol{x}}_t^{(k-1)} \right\|^2 = \left\| \boldsymbol{z}_{-K+k-2} - f_{k-1}([\hat{\boldsymbol{x}}_{t-1}^{(k)},\ldots,\hat{\boldsymbol{x}}_{t-1}^{(K)},0,\ldots,0],\boldsymbol{z}_t) \right\|^2$$

$$\leq 2 \left\| \boldsymbol{z}_{-K+k-2} - f_{k-1}([\boldsymbol{z}_{-K+k-2},\hat{\boldsymbol{x}}_{t-1}^{(k+1)},\ldots,\hat{\boldsymbol{x}}_{t-1}^{(K)},0,\ldots,0],\boldsymbol{z}_t) \right\|^2$$

$$+ 2 \left\| f_{k-1}([\boldsymbol{z}_{-K+k+t-1},\hat{\boldsymbol{x}}_{t-1}^{(k+1)},\ldots,\hat{\boldsymbol{x}}_{t-1}^{(K)},0,\ldots,0],\boldsymbol{z}_t) - f_{k-1}([\hat{\boldsymbol{x}}_{-2}^{(k)},\ldots,\hat{\boldsymbol{x}}_{t-1}^{(K)},0,\ldots,0],\boldsymbol{z}_t) \right\|^2$$

$$\leq 2\frac{\varepsilon^2}{C_G^2} + 2L \left\| \boldsymbol{z}_{-K+k+t-1} - \hat{\boldsymbol{x}}_{t-1}^{(k)} \right\|^2 \leq 2\frac{\varepsilon^2}{C_G^2} + \sum_{j=1}^{K-k} (2L)^{j+1} \frac{\varepsilon^2}{C_G^2} \leq \sum_{j=1}^{K-k+1} (2L)^j \frac{\varepsilon^2}{C_G^2},$$

which completes the induction. Therefore, we obtain

$$\left\|(\boldsymbol{z}_{-K+1},\ldots,\boldsymbol{z}_0) - ([\hat{\boldsymbol{x}}_{-1}^{(2)},\ldots,\hat{\boldsymbol{x}}_{-1}^{(K)}],\boldsymbol{z}_0)\right\|^2 = \sum_{k=2}^{K}\left\|\boldsymbol{z}_{-K+k-1} - \hat{\boldsymbol{x}}_{-1}^{(k)}\right\|^2$$

$$\leq \sum_{k=2}^{K}\left\|\boldsymbol{z}_{-K+k-1} - \hat{\boldsymbol{x}}_{-1}^{(k)}\right\|^2 \leq \frac{\varepsilon^2}{C_G^2}\sum_{k=2}^{K}\sum_{j=1}^{K-k+1}(2L)^j = \frac{\varepsilon^2}{16L_G^2}$$

From (43), we thus obtain

$$
\sup_{\boldsymbol{z}\in(D_d)^{K+1}}\left\|G(\boldsymbol{z}_{-K+1},\ldots,\boldsymbol{z}_0) - \hat{\boldsymbol{x}}_0^{(1)}\right\|
$$
$$
\leq L_G\left\|(\boldsymbol{z}_{-K+1},\ldots,\boldsymbol{z}_0) - ([\hat{\boldsymbol{x}}_{-1}^{(2)},\ldots,\hat{\boldsymbol{x}}_{-1}^{(K)}],\boldsymbol{z}_0)\right\| + \frac{\varepsilon}{4} \leq \frac{\varepsilon}{2}. \tag{44}
$$

Setting $W$ to be the projection onto the first block $\hat{\boldsymbol{x}}_0^{(1)}$, (that is, $W$ has zero entries except for $W_{i,i} = 1$ for $i = 1,\ldots,m$) and putting together (36), (37) and (44) yields

$$
\sup_{\boldsymbol{z}\in(D_d)^{\mathbb{Z}_-}}\sup_{t\in\mathbb{Z}_-}\left\|H_U(\boldsymbol{z}) - H_{\bar{U}_W}(\boldsymbol{z})\right\| \leq \sup_{\boldsymbol{z}\in(D_d)^{\mathbb{Z}_-}}\left\|H_U(\boldsymbol{z}) - \bar{G}(\boldsymbol{z}_{-K+1},\ldots,\boldsymbol{z}_0)\right\|
$$
$$
+ \sup_{\boldsymbol{z}\in(\mathbb{R}^d)^K}\left\|G(\boldsymbol{z}) - \bar{G}(\boldsymbol{z})\right\| + \sup_{\boldsymbol{z}\in(D_d)^{K+1}}\left\|G(\boldsymbol{z}_{-K+1},\ldots,\boldsymbol{z}_0) - H_{\bar{U}_W}(\boldsymbol{z})\right\| \tag{45}
$$
$$
\leq \frac{\varepsilon}{4} + \frac{\varepsilon}{4} + \frac{\varepsilon}{2} = \varepsilon.
$$

It remains to be shown that (14) has the echo state property. Recall that we partition $\hat{\boldsymbol{x}}_t = [\hat{\boldsymbol{x}}_t^{(1)},\ldots,\hat{\boldsymbol{x}}_t^{(K)}] \in \mathbb{R}^m \times (\mathbb{R}^d)^{K-1}$. For $k = 1$, $j \in \{1,\ldots,l_1\}$, and $k = 2,\ldots,K-1$, $j \in \{l_{k-1}+1,\ldots,l_k\}$, select $P_j$ as the matrix with zero entries, except for $(P_j)_{l,l+l_k} = 1$ for $l = 1,\ldots,d(K-k)$ and let $P_j = 0$ for $j = l_{K-1}+1,\ldots,N$. Then, for $k = 1$, $j \in \{1,\ldots,l_1\}$, and $k = 2,\ldots,K-1$, $j \in \{l_{k-1}+1,\ldots,l_k\}$, we have $P_j\hat{\boldsymbol{x}}_t = [\hat{\boldsymbol{x}}_t^{(k+1)},\ldots,\hat{\boldsymbol{x}}_t^{(K)},0,\ldots,0]$ and $P_j\hat{\boldsymbol{x}}_t = 0$ for $j = l_{K-1}+1,\ldots,N$. Then, echo state property follows by calling Lemma 4.7. Therefore, the approximation bound for the functional (45) immediately implies the corresponding bound for the filter (17), which completes the proof of the theorem.

$\square$

## D  CONSTRUCTION OF V

In this appendix we provide further details on the choice of $V$ appearing in the quantum circuit. Our presentation follows Gonon & Jacquier (2025).

Generally, the matrix $V \in \mathbb{C}^{n_U \times n_U}$ can be any unitary matrix mapping $|0\rangle^{\otimes n}$ to the state $|\psi\rangle = \frac{1}{\sqrt{n}}\sum_{i=0}^{n-1}|4i\rangle$ which, for $n \geq 2$, is also explicitly given as $|\psi\rangle = \frac{1}{\sqrt{n}}\sum_{i=0}^{n-1}|i\rangle \otimes |00\rangle$.

As $V|0\rangle^{\otimes n} = |\psi\rangle$ is the only property required in the proof, many alternative choices of $V$ are possible and one may thus select the one that is most suitable from the perspective of hardware requirements or limitations.

**Example**  One explicit example for $V$ is given by $V := 2|\varphi\rangle\langle\varphi| - I$, with

$$|\varphi\rangle := \frac{|0\rangle + |\psi\rangle}{\sqrt{2(1 + \langle 0|\psi\rangle)}},$$

where we write $|0\rangle$ in place of $|0\rangle^{\otimes n}$ for brevity here. One easily checks that $V^\dagger = 2|\varphi\rangle\langle\varphi| - I = V$ and thus $VV^\dagger = V^\dagger V = I$. Furthermore, a straightforward computation yields that

$$V|0\rangle = (2|\varphi\rangle\langle\varphi| - I)|0\rangle$$
$$= \frac{|0\rangle(1 + \langle\psi|0\rangle) + |\psi\rangle(1 + \langle\psi|0\rangle)}{1 + \langle 0|\psi\rangle} - |0\rangle = |\psi\rangle.$$

**Construction of $|\psi\rangle$**  In the case $n_0 = 0$, there is an explicit construction of $\psi$ in terms of Hadamard gates acting on the control qubits. Indeed, for $\mathfrak{n} \geq 2$, (Gonon & Jacquier, 2025, Lemma A.2) shows that

$$|\psi_l\rangle_{\mathfrak{n}_l} = \left(\bigotimes_{i=0}^{\mathfrak{n}_l-2} \mathrm{H}\,|0\rangle\right) \otimes |00\rangle\,.$$

# E   Monte Carlo Error

In practice, the empirical sampling error leads to an additional error component of order $1/\sqrt{S}$ for $S$ independent shots, see, e.g., Qi et al. (2023); Liu et al. (2025). Here, we outline how this Monte Carlo error could be taken into account in the present setting.

More specifically, our QNNs in (3) and (4) are defined using probabilities, rather than their Monte Carlo estimates

$$\widehat{\mathbb{P}}_m^{n,\boldsymbol{\theta}} := \frac{1}{S} \sum_{s=1}^{S} \mathbb{1}_{\{m, 4+m, \ldots, 4(n-1)+m\}}(i^{(s)}),$$

with $i^{(s)}$ the measured state in the $i$-th shot. To obtain refined bounds incorporating the sampling error, one would proceed as follows. Denote by $\bar{F}_R^{n,\boldsymbol{\theta},S}$ the RQNN state map with output probabilities estimated by $S$ shots, by $\hat{\boldsymbol{x}}^S$ the associated state and by $\bar{U}_S$ the associated filter.

For the state map itself, the $L^2$-error can be directly controlled (as in Gonon & Jacquier (2025)) by

$$\mathbb{E}\left[\int_{\mathbb{R}^N \times \mathbb{R}^d} \left|\bar{F}_{R,j}^{n,\boldsymbol{\theta}}(\boldsymbol{x},\boldsymbol{z}) - \bar{F}_{R,j}^{n,\boldsymbol{\theta},S}(\boldsymbol{x},\boldsymbol{z})\right|^2 \mu(\mathrm{d}\boldsymbol{x}, \mathrm{d}\boldsymbol{z})\right]^{1/2}$$

$$\leq 2R \sum_{i=1}^{2} \left(\int_{\mathbb{R}^N \times \mathbb{R}^d} \mathbb{E}\left[\left|\mathbb{P}_i^{n,\boldsymbol{\theta}^j}(\boldsymbol{x},\boldsymbol{z}) - \widehat{\mathbb{P}}_i^{n,\boldsymbol{\theta}^j}(\boldsymbol{x},\boldsymbol{z})\right|^2\right] \mu(\mathrm{d}\boldsymbol{x}, \mathrm{d}\boldsymbol{z})\right)^{1/2} \qquad (46)$$

$$\leq \frac{4R}{\sqrt{S}},$$

using that $\mathbb{E}[|\mathbb{E}[X_1] - \frac{1}{S}\sum_{s=1}^{S} X_s|^2] = \frac{\mathrm{Var}(X_1)}{S}$ for i.i.d. random variables $X_1, \ldots, X_S$.

For the associated filter, one may proceed as follows. Firstly, (33) in the proof of Theorem 4.6 can be adapted to

$$\left\|\bar{U}_S(\boldsymbol{z})_t - U(\boldsymbol{z})_t\right\| = \left\|\hat{\boldsymbol{x}}_t^S - \boldsymbol{x}_t\right\| = \left\|\bar{F}_R^{n,\boldsymbol{\theta},S}(\hat{\boldsymbol{x}}_{t-1}^S, \boldsymbol{z}_t) - F(\boldsymbol{x}_{t-1}, \boldsymbol{z}_t)\right\|$$

$$\leq \left\|F(\boldsymbol{x}_{t-1}, \boldsymbol{z}_t) - F(\hat{\boldsymbol{x}}_{t-1}^S, \boldsymbol{z}_t)\right\| + \left\|F(\hat{\boldsymbol{x}}_{t-1}^S, \boldsymbol{z}_t) - \bar{F}_R^{n,\boldsymbol{\theta},S}(\hat{\boldsymbol{x}}_{t-1}^S, \boldsymbol{z}_t)\right\|$$

$$\leq \lambda \left\|\boldsymbol{x}_{t-1} - \hat{\boldsymbol{x}}_{t-1}^S\right\| + \left(\sum_{j=1}^{N} \left\|\bar{F}_{R,j}^{n,\boldsymbol{\theta},S} - \bar{F}_{R,j}^{n,\boldsymbol{\theta}}\right\|_{\infty,M}^2\right)^{1/2} + \left(\sum_{j=1}^{N} \left\|\bar{F}_{R,j}^{n,\boldsymbol{\theta}} - F_j\right\|_{\infty,M}^2\right)^{1/2} \qquad (47)$$

$$\leq \lambda \left\|\boldsymbol{x}_{t-1} - \hat{\boldsymbol{x}}_{t-1}^S\right\| + \frac{\sqrt{N} \max_{j=1,\ldots,N} C_j^\infty}{\sqrt{n}} + \left(\sum_{j=1}^{N} \left\|\bar{F}_{R,j}^{n,\boldsymbol{\theta},S} - \bar{F}_{R,j}^{n,\boldsymbol{\theta}}\right\|_{\infty,M}^2\right)^{1/2}.$$

The last error term can be bounded as

$$\mathbb{E}\left[\left(\sum_{j=1}^{N} \left\|\bar{F}_{R,j}^{n,\boldsymbol{\theta},S} - \bar{F}_{R,j}^{n,\boldsymbol{\theta}}\right\|_{\infty,M}^2\right)^{1/2}\right] \leq \left(\sum_{j=1}^{N} \mathbb{E}\left[\left\|\bar{F}_{R,j}^{n,\boldsymbol{\theta},S} - \bar{F}_{R,j}^{n,\boldsymbol{\theta}}\right\|_{\infty,M}^2\right]\right)^{1/2} \leq \frac{C}{\sqrt{S}}$$

for a suitable constant $C$ using techniques from statistical learning theory, provided that $\widehat{\mathbb{P}}_m^{n,\boldsymbol{\theta}}$ is Lipschitz continuous as a function of $(\boldsymbol{x}, \boldsymbol{z})$. Inserting this into (47) and proceeding precisely as in the proof of Theorem 4.6 then yields a bound that incorporates also the sampling error.

Alternatively, as the Lipschitz continuity may be hard to verify, we may obtain an $L^2$-bound analogously to Theorem 4.6 as follows. First, using that the shots are independent across evaluations, we may apply (46) to estimate

$$\mathbb{E}\left[\left(\sum_{j=1}^{N}\left\|\bar{F}_{R,j}^{n,\boldsymbol{\theta},S}(\hat{\boldsymbol{x}}_{t-1}^{S},\boldsymbol{z}_t)-\bar{F}_{R,j}^{n,\boldsymbol{\theta}}(\hat{\boldsymbol{x}}_{t-1}^{S},\boldsymbol{z}_t)\right\|^2\right)^{1/2}\right]$$

$$\leq\left(\sum_{j=1}^{N}\mathbb{E}\left[\left\|\bar{F}_{R,j}^{n,\boldsymbol{\theta},S}(\hat{\boldsymbol{x}}_{t-1}^{S},\boldsymbol{z}_t)-\bar{F}_{R,j}^{n,\boldsymbol{\theta}}(\hat{\boldsymbol{x}}_{t-1}^{S},\boldsymbol{z}_t)\right\|^2\right]\right)^{1/2}$$

$$\leq\sqrt{N}\frac{4R}{\sqrt{S}},$$

where the expectations are taken with respect to sampling the probabilities to evaluate $\bar{F}_{R,j}^{n,\boldsymbol{\theta},S}(\hat{\boldsymbol{x}}_{t-1}^{S},\boldsymbol{z}_t)$.

Next, by proceeding as in (47), we may estimate

$$\mathbb{E}[\|\bar{U}_S(\boldsymbol{z})_t-U(\boldsymbol{z})_t\|]\leq\lambda\|\boldsymbol{x}_{t-1}-\hat{\boldsymbol{x}}_{t-1}^{S}\|+\frac{\sqrt{N}\max_{j=1,\ldots,N}C_j^\infty}{\sqrt{n}}$$

$$+\mathbb{E}\left[\left(\sum_{j=1}^{N}\left\|\bar{F}_{R,j}^{n,\boldsymbol{\theta},S}(\hat{\boldsymbol{x}}_{t-1}^{S},\boldsymbol{z}_t)-\bar{F}_{R,j}^{n,\boldsymbol{\theta}}(\hat{\boldsymbol{x}}_{t-1}^{S},\boldsymbol{z}_t)\right\|^2\right)^{1/2}\right] \tag{48}$$

$$\leq\lambda\|\boldsymbol{x}_{t-1}-\hat{\boldsymbol{x}}_{t-1}^{S}\|+\frac{\sqrt{N}\max_{j=1,\ldots,N}C_j^\infty}{\sqrt{n}}+\sqrt{N}\frac{4R}{\sqrt{S}}$$

with the expectations again taken with respect to sampling the probabilities to evaluate $\bar{F}_{R,j}^{n,\boldsymbol{\theta},S}(\hat{\boldsymbol{x}}_{t-1}^{S},\boldsymbol{z}_t)$. In particular, taking expectations also with respect to a random process $\mathbf{Z}$ (taking values in $(D_d)^{\mathbb{Z}_-}$) and sampling at each evaluation, the estimate (48) and the same arguments as in the proof of Theorem 4.6 yield the bound

$$\sup_{t\in\mathbb{Z}_-}\mathbb{E}[\|U^F(\mathbf{Z})_t-\bar{U}_S(\mathbf{Z})_t\|]\leq\frac{1}{1-\lambda}\left(\frac{\sqrt{N}\max_{j=1,\ldots,N}C_j^\infty}{\sqrt{n}}+\sqrt{N}\frac{4R}{\sqrt{S}}\right). \tag{49}$$

