# OpenReview forum: "Feedback-driven recurrent quantum neural network universality"
_ICLR.cc/2026/Conference — ICLR 2026 Poster_

### Official Review · Reviewer_WKBd · 2025-10-29

**Soundness:** 3
**Presentation:** 3
**Contribution:** 3
**Rating:** 6
**Confidence:** 4

**Summary:**

This theoretical paper discusses the universal approximation bound for RQNNs. The authors extend approximation error bounds from previous QNN work and introduce a time factor as a state-space system to derive the approximation bound for RQNNs.

**Strengths:**

1. The paper is well-written and logically structured.
2. The subject is important for the quantum reservoir computing community, providing theoretical foundations for a field dominated by empirical studies.

**Weaknesses:**

The presentation could be more reader-friendly with graphical illustrations of the problem statement and derivation direction (nice to have but minor).

**Questions:**

1. In equation (12), where N = 2^{qubit count} relates to qubit size and n relates to parameter count, the role of λ (the gradient bound) in the denominator appears extremely important. How would λ be determined in practice? For example, does it relate to quantum noise?
2. Can the measurement shot requirement for extracting quantum information from the QRNN play an important role in the approximation bound? For example, the empirical estimation of expectation value typically has a distance error of the form O(1/√N_{shot}) (ref. eq. (19) in [1] and also eq. (19) in [2]).

[1] Qi, J., Yang, CH.H., Chen, PY. et al. Theoretical error performance analysis for variational quantum circuit based functional regression. npj Quantum Inf 9, 4 (2023)

[2] Liu, CY., Kuo, EJ., Abraham Lin, CH. et al. Quantum-Train: rethinking hybrid quantum-classical machine learning in the model compression perspective. Quantum Mach. Intell. 7, 80 (2025).

---

> ### Author Response · Authors · 2025-11-19
>
> Dear reviewer,
>
> Thank you very much for your time and effort in reviewing our paper and for your fruitful suggestions. We are very happy to read your positive assessment.
>
> **Weaknesses**
>
> >The presentation could be more reader-friendly with graphical illustrations of the problem statement and derivation direction (nice to have but minor).
>
>  We thank the reviewer for pointing this out. We have added a new figure of the circuit scheme and a paragraph at the end of Section 3 to help the reader in understanding the algorithm. We have also added several sentences in various places to improve the readability and presentation of the paper.
>
> **Questions**
>
> > In equation (12), where N = 2^{qubit count} relates to qubit size and n relates to parameter count, the role of λ (the gradient bound) in the denominator appears extremely important. How would λ be determined in practice? For example, does it relate to quantum noise?
>
> We thank the reviewer for this comment, we realized that the previous presentation was not transparent enough about the role of λ. We have modified the statement and proof of Theorem 4.6 to explicitly display λ as the maximum Lipschitz constant of the target functional. This means that we are delimiting the range of targets that we can explicitly approximate with Eq.(12) to Lipschitz contracting functions, where λ is the maximum rate of contraction. This condition gathers both smoothness conditions and the echo state property for the target, which are fundamental for the proof. We would like to clarify that to use Theorem 4.6, we need to make sure that λ exists, but it does not need to be determined in practice. Given 0<λ<1 and N (target dimension), Eq.(12) ensures that we can control the approximation error by increasing n. In fact, the specific value of λ is not related to the properties of the QRNN, since for large enough n, the echo state property of the QRNN is guaranteed.
> We have also realized that N = 2^{qubit count} was a typo, since N represents the state space dimension, which is independent of the Hilbert space dimension of the quantum circuit. Instead, we have introduced the variable $n_\mathrm{U}$ at the beginning of Section 3 as the dimension of the Hilbert space. We now clarify that N is both the target and QRNN state space dimension in Theorem 4.6, while it is only the QRNN state space dimension in Theorem 4.8.
>
> > Can the measurement shot requirement for extracting quantum information from the QRNN play an important role in the approximation bound? For example, the empirical estimation of expectation value typically has a distance error of the form O(1/√N_{shot}) (ref. eq. (19) in [1] and also eq. (19) in [2]).
>
>
> Thanks for pointing out this important point. The shot noise error is an additional error component that we did not consider in the error bounds, since the main challenge comes from the RQNN approximation in the temporal domain. To discuss in detail the effects of shot noise,  we added a new Appendix E where we outline in detail how the shot noise can be taken into account in our mathematical framework and mentioning both references that you provided.
>
> We hope that these clarifications and adjustments have helped to address all raised questions.

---

> > ### Comment · Reviewer_WKBd · 2025-11-20
> >
> > The authors have addressed all my questions, so I raised the rating.

---

### Official Review · Reviewer_mUbU · 2025-10-31

**Soundness:** 4
**Presentation:** 4
**Contribution:** 3
**Rating:** 8
**Confidence:** 3

**Summary:**

The paper derives approximation bounds and universality statements for recurrent quantum neural networks. The proposed approach is based on a uniformly controlled quantum gate to apply multicontrolled rotations to a set of control and target qubits, and it has been recently shown that it can be efficiently implemented.

The authors first prove that RQNNs are able to uniformly approximate the filters induced by any contracting Barrontype state-space system.
Second, they extend this universality property to the much larger category of arbitrary fading memory, causal, and time-invariant filters. In this last result, neither Barron-type integrability nor contractivity conditions are needed for the target filter.

The paper is a strong theoretical contribution to the field. One of its major limitations for ICLR is that it is only theoretical.

**Strengths:**

- Relevant theoretetical contribution
- Excellent technical depth and rigor
- Clear positioning in literature and in particular w.r.t recent literature

**Weaknesses:**

- Lack of empirical validation. Theoretical findings are strong. However, no numerical or experimental results are a limitation for this paper.
- Some assumption (e.g., Barron-type integrability) may restrict practical applicability
- A comparison between the proposed approach and classical RNNs or RC models for large n

**Questions:**

Could you discuss the fact that error rates do not suffer from the curse of dimensionality? I think you refer to d, but  I'm not sure.

---

> ### Author Response · Authors · 2025-11-19
>
> Dear reviewer,
>
> Thank you very much for your time and effort in reviewing our paper and for your fruitful suggestions. We are very happy to read your positive assessment.
>
> **Weaknessess**
>
> > Lack of empirical validation. Theoretical findings are strong. However, no numerical or experimental results are a limitation for this paper.
>
> Our work is indeed primarily theoretical, and its core contribution is to establish new analytical bounds and to clarify their behavior in comparison with existing theoretical guarantees in the literature (see also the answer to the question about comparison to classical RNNs below). A variety of feedback architectures have already been analysed experimentally. In this context, the primary objective of our work is not empirical performance but rather the advancement of the theoretical understanding of the problem. We therefore focus on deriving bounds that can be directly compared, on equal footing, with prior theoretical results. We believe this comparison highlights the significance of our contribution and is the appropriate standard of evaluation for the type of result we provide. In that direction, we have enhanced our presentation with a paragraph, right under Theorem 4.6, that emphasizes crucial advantages of QRNNs when compared to classical RNN counterparts where we point out, for instance, that our results show that the regularity requirements on the target systems are *strictly weaker* using QRNNs as opposed to classical RNNs. Having said all this, numerical evaluation experiments are part of our research agenda and will be executed in future work.
>
> > Some assumption (e.g., Barron-type integrability) may restrict practical applicability
>
> We thank the reviewer for raising the concern about the practicality of Barron-type integrability assumptions. We agree that such conditions may not hold for every real-world application; however, they are a widely used and well-established theoretical device for understanding the representational and generalization properties of neural learning paradigms. Many influential works use Barron-type spaces precisely because they allow for interpretable bounds while still capturing a broad and practically relevant class of functions. Our goal is to advance theoretical understanding within this standard framework, and the assumptions we adopt are consistent with prior foundational analyses. Moreover, these integrability conditions often serve as sufficient, not necessary, conditions: in practice, networks can perform well even outside strict Barron-space regimes, but the theory currently relies on these assumptions to produce meaningful guarantees. Thus, rather than limiting applicability, the Barron framework provides a recognized and informative baseline for comparing theoretical results across studies as we do, for example, in the answer to the comment below.
>
> > A comparison between the proposed approach and classical RNNs or RC models for large n
>
>
> Thanks a lot for this comment. We agree that it is important to compare to classical RNN counterparts. Indeed, the most similar setup with classical RNN has been considered in Theorem 3 of [Gonon, Grigoryeva, Ortega 2023]. While the approximation rates are the same in both results, the QRNNs offer the crucial advantage that the smoothness requirements are less stringent. In other words, the regularity requirements on the target systems are *strictly weaker* using QRNNs as opposed to classical RNNs. For example, for Sobolev function approximation classical RNNs require Sobolev regularity $N+d+3$ while QRNNs only requires $\frac{N+d}{2}+4$. While this is not a quantum advantage in the classical sense, this result still shows that QRNN provide an advantage in the sense that they are able to approximate a broader class of state-space systems.
>
> **Questions**
>
> > Could you discuss the fact that error rates do not suffer from the curse of dimensionality? I think you refer to d, but I'm not sure.
>
> Yes, this is correct. We now make this more explicit by saying *In particular, this result shows that the error rate is free from the curse of dimensionality: the error decays as $\frac{1}{\sqrt{n}}$ as we increase $n$, with this rate of decay being independent of the input dimension $d$ and the state space dimension $N$.*
>
> We hope that these clarifications and adjustments have helped to address all raised questions.

---

### Official Review · Reviewer_7EQ4 · 2025-10-31

**Soundness:** 4
**Presentation:** 2
**Contribution:** 4
**Rating:** 8
**Confidence:** 3

**Summary:**

Summary

The manuscript rigorously derives approximation error bounds and a
proof for universality of recurrent quantum neural networks (RQNNs).
The main novelty is the inclusion of recurrency/feedback, as similar
results have been recently presented for feedforward quantum neural
networks (Gonon & Jacquier 2025). Universal approximation theorems are
an important topic and well researched for classical neural networks,
as they provide important insights on the power of different neural
network architectures. Given the importance of recurrent networks in
processing time series, the results of this study are highly relevant.


Soundness

The study is purely theoretical. The obtained results, however, look
sound as detailed proofs are presented in the appendices. I admit that
I did not dive deeply into all aspects of the proofs, but many aspects
seem to rely on closely related previous studies on feedforward QNNs.


Presentation

The general structure of the study is very good. The main text
focuses on the main results, and detailed proofs are being relegated
to appendix sections. Also the appendix on previous techniques in
quantum neural networks is very helpful.

The authors further attempt to provide simple introductions to the
individual sections to guide the reader through the logic of the
construction of the quantum circuits and the proofs for their
approximation capabilities. While this helped me to a certain extent,
in my opinion, the manuscript is still hard to follow for non-experts
on quantum neural networks. Given that the ICLR community has very
broad backgrounds, publication at ICLR therefore requires some
improvements in terms of presentation. The main issue I have is that
the manuscript is not self-contained enough. See detailed points in
weaknesses below.


Contribution

The authors present a systematic way to construct unitary operators
from a suitable combination of rotation and Hadamard gates that upon
measurement after application to initial states define a set of
functions that can approximate target activation functions of neurons
in recurrent networks to arbitrary accuracy. This is done by tuning
the parameters of the gates. Mathematically rigorous proofs are
presented for these construction steps and the resulting approximation
power. The analysis largely relies on a recently published study on
similar error bounds and universality properties of quantum
feedforward neural networks. For non-expert readers, the novelty, i.e.
the specific differences in the derivations that feedback and recurrency
introduce, are hard to detect and not clearly presented enough.

**Strengths:**

The derivations in the study seem mathematically rigorous. The authors
also provide useful background on filters and functionals, and a
helpful review of existing literature in the related works and
appendix sections.

**Weaknesses:**

The study is not self-contained enough:
references in many places to previous work Gonon and Jacquier (2025).
This makes it harder to follow. In particular, this issue appears when
the authors aim to introduce the unitary matrix V, which seems to be
important for the recurrent quantum neural network architecture. But
all details are relayed to the previous publication.

The study needs to better work out the novelty:
As often mentioned many times throughout the manuscript, the work
largely follows the techniques in Gonon & Jacquier 2025 and others on
feedforward QNNs. The differences between the feedforward and the
recurrent case of QNNs, however, needs to be highlighted more strongly
so that the novelty of the results also becomes more apparent to
readers that are not experts in the field and familar with the
previous studies. In particular concerning the constructions in
Section 3: how do they differ explicitly from the case of feedforward
networks? This is not obvious, but crucial to judge the advances
compared to Gonon & Jacquier 2025.

Proposition 4.1:
This proposition seems to be central for the understanding of the
procedure. The proof of proposition 4.1 is only relayed to Gonon &
Jacquier 2025. For a more self-contained presentation that targets the
broader community of ICLR, it would be better to also present the
proof of proposition 4.1 in the appendix of the current study.

The "curse of dimensionality" is emphasized in the abstract,
introduction and the summary of contributions, but never mentioned in
the results sections. It would be helpful to point to the results on
dimensionality more specifically throughout the manuscript and
emphasize the importance on the log scaling there.

**Questions:**

Reservoir computing typically optimizes only the readout layer. The
authors mention in Section 1.2. that their results are for networks
where all parameters are trainable, but they claim that they are also
generalizable to random parameters in the recurrent layer. This
generalization does not become clear from the current presentation.
This point is only mentioned again in the conclusion section, but it
is not clear how the analyses of the current study support this claim.
This point needs to be elaborated much further.


Minor points:
- Clarify why it is necessary to extend previous results on functions to the first derivatives.
- The abbreviation SAS is not defined.
- Please spell out Barron-type conditions somewhere.
- Please elaborate more on the function of control and target qubits for non-expert readers.
- Please make clearer around line 266 that one needs N parallel circuits to have one for approximating each component of F in eq. (1)?

---

> ### Author Response · Authors · 2025-11-19
>
> Dear reviewer,
>
> Thank you very much for your time and effort in reviewing our paper and for your fruitful suggestions. We are very happy to read your positive assessment.
>
> **Weaknessess**
>
> > The study is not self-contained enough: ...  In particular, this issue appears when the authors aim to introduce the unitary matrix V, which seems to be important for the recurrent quantum neural network architecture. But all details are relayed to the previous publication.
>
> Thank you for raising this point. We have made the presentation clearer by emphasizing in a newly introduced paragraph in page 5 that many different choices of V are possible. Additionally, we have created a new Appendix (Appendix D) that provides further details on the choice of V appearing in the quantum circuit and contains an example choice. We hope that this makes the presentation clearer.
>
>
> > ...  In particular concerning the constructions in Section 3: how do they differ explicitly from the case of feedforward networks? This is not obvious, but crucial to judge the advances compared to Gonon & Jacquier 2025.
>
> Thank you, we agree that it is important to better state the novelty of our work. This paper extends the aproximation bounds and universality results of the static framework in Gonon & Jacquier 2025 to the temporal domain.  To do so, we mainly add two new ingredients: 1) We introduce the state vector $x$ into the argument of the QNN as incoherent feedback to provide memory to the reservoir; 2) we derive new QNN approximation results that allow to jointly approximate functions and their derivatives, see  Proposition 4.4 and Corollary 4.5, respectively. These results are useful on their own, but also provide key ingredients for proving the approximation and universality Theorems 4.6 and 4.8.
> The temporal domain is inherently much more challenging due to the feedback loop, and hence, proving Theorems 4.6 and 4.8 required to develop new techniques specifically tailored to deal with this situation (see Appendix C).
> We have added sentences in the Introduction (Contributions) and Section 3 and 4 to clarify these aspects.
>
> > Proposition 4.1: This proposition seems to be central for the understanding of the procedure. The proof of proposition 4.1 is only relayed to Gonon & Jacquier 2025. For a more self-contained presentation that targets the broader community of ICLR, it would be better to also present the proof of proposition 4.1 in the appendix of the current study.
>
> Thank you, we agree that it would be helpful to add the detailed proof. Therefore, we have added a complete proof of Proposition 4.1 to Appendix B.
>
> > The "curse of dimensionality" is emphasized in the abstract, introduction and the summary of contributions, but never mentioned in the results sections. It would be helpful to point to the results on dimensionality more specifically throughout the manuscript and emphasize the importance on the log scaling there.
>
> Thank you very much for pointing this out. We have now made this more explict both at the beginning of the results section and also before the approximation result.

---

> > ### Author Response · Authors · 2025-11-19
> >
> > **Questions**
> >
> > > Reservoir computing typically optimizes only the readout layer. The authors mention in Section 1.2. that their results are for networks where all parameters are trainable, but they claim that they are also generalizable to random parameters in the recurrent layer. This generalization does not become clear from the current presentation. This point is only mentioned again in the conclusion section, but it is not clear how the analyses of the current study support this claim. This point needs to be elaborated much further.
> >
> > Thank you for pointing this out. Our intention here was to point out that the techniques developed here may also be useful for further studies, e.g., on randomized architectures and studies of the generalization error.  We have adjusted the wording to clarify that this connection has not been made yet, but will require further analysis; thereby we consider it a fruitful direction of future research.
> >
> > **Minor points**:
> > >- Clarify why it is necessary to extend previous results on functions to the first derivatives.
> >
> > The extension to the first derivatives is a refinement  of the results in Gonon & Jacquier 2025 under similar Barron-type integrability conditions. The need for such a statement is the dynamical character of the statement that is proved; the goal is not only approximating a state-space equation but, more importantly, the filter that it generates (in the presence of the echo state property). More specifically, having control on the quality of the approximation of the derivatives of the state-space map is what, later on in in the proofs of Theorems 4.6 and 4.8, allows for the transfer of this level of approximation to the corresponding filters (in the presence of bound conditions on the norms of the gradients of the systems that need to be approximated).
> >
> > >-    The abbreviation SAS is not defined.
> >
> > SAS stands for state-affine systems. This has been clarified in the text.
> >
> > >-    Please spell out Barron-type conditions somewhere.
> >
> > Explicit examples of Barron-type integrability conditions have been spelled out in the introductory paragraph to Section 4.
> >
> > >-    Please elaborate more on the function of control and target qubits for non-expert readers.
> >
> > In the quantum computing literature, the control qubits are the inputs of the controlled quantum gates, which only modify the state of the target qubits. The most simple example is the CNOT gate, which is the quantum version of the classical XOR gate for two qubits. That is, if {0,1} are the only allowed values for the target and control qubit, the CNOT gate acts as the addition modulo 2 over the target qubit state. Then, multi-controlled unitaries are a generalization of the CNOT gate where one can have several control and target qubits. We have added a similar explanation in the main text.
> >
> > >-    Please make clearer around line 266 that one needs N parallel circuits to have one for approximating each component of F in eq. (1)?
> >
> > Thanks for the observation. A clarification has been added.
> >
> >
> > We hope that these clarifications and adjustments have helped to address all raised questions.

---

> > > ### Comment · Reviewer_7EQ4 · 2025-11-25
> > >
> > > I thank the authors for taking so much care in addressing the points of presentation raised in my review. I keep my score and would be pleased to see this work being presented at the conference.

---

> > > > ### Author Response · Authors · 2025-11-25
> > > >
> > > > We are very happy to hear that all your concerns and questions have been clarified. Thank you very much once more for all the valuable feedback and the encouraging comments.

---

### Official Review · Reviewer_A7sM · 2025-11-01

**Soundness:** 4
**Presentation:** 4
**Contribution:** 4
**Rating:** 10
**Confidence:** 4

**Summary:**

This paper investigates the expressive power of recurrent quantum neural networks (RQNNs) in processing temporal data. The authors address a significant gap in quantum machine learning theory – namely, whether quantum recurrent models (a form of quantum reservoir computing with feedback) can universally approximate sequences and dynamical systems, and if so, under what resource requirements. They develop a rigorous theoretical framework combining quantum neural network function approximation results with classical reservoir computing theory. The main contributions include quantitative approximation error bounds and universality theorems for RQNNs with simple linear output layers.

**Strengths:**

1. IMHO, this work is the first to establish quantitative universal approximation bounds for recurrent quantum neural networks. Prior to this, the literature lacked error guarantees for quantum recurrent models. The paper fills that gap by providing rigorous theorems (with proofs) that demonstrate RQNNs’ ability to approximate a broad class of time-dependent functions to arbitrary accuracy.
2. Also, it shows that RQNNs can achieve universality without the need for high-degree polynomial readout functions.
3. to reach a desired approximation error, the required number of qubits grows only logarithmically with $1/\varepsilon$. In other words, exponentially increasing the accuracy only adds a linear number of additional qubits. This is a remarkable claim as it suggests no curse of dimensionality in qubit resources.

**Weaknesses:**

1. Thm 4.6 relies on the requirement that the state transition function lies in the Barron function class and has bounded first derivatives (plus contractivity $\lambda<1$), which means the results apply primarily to “well-behaved” systems (smooth, band-limited, and not too chaotic). Real-world temporal processes might violate these conditions (e.g., non-smooth or highly non-contractive dynamics).
2. Minor weakness: the paper does not include any experimental or numerical simulation results to complement the theory. All results are analytical. This is a weakness in the context of a machine learning conference. Still, what I said is merely a comment, rather than a criticism.
3. The paper assumes the existence of optimal parameters $\theta$ for the RQNN (since it’s a universal approximation argument), but does not discuss how one might find these parameters in practice. Training a quantum model with many parameters is non-trivial, e.g., you might face barren plateaus, circuit noises, etc. You can argue that this is beyond the scope of this paper. Still, I think it is quite an important aspect to at least discuss them in the conclusion.

**Questions:**

1. The paper asserts that RQNNs have approximation capabilities “as competitive as” classical reservoir families like echo state networks or state-affine systems. However, it doesn’t provide a direct comparison or quantification of any potential advantage. Hmm.. IMHO, most ppl in QML would ask about the potential for quantum advantage here, for instance, do there exist learning instances such that RQNNs and their classical counterparts exhibit a learning separation in terms of either time or sample complexity?
2. Have the authors considered the practical side of how one would train or set the parameters $\theta$ for an RQNN to approximate a given system? Would one use gradient-based variational quantum circuits training to fit observed data from the target system?

---

> ### Author Response · Authors · 2025-11-19
>
> Dear reviewer,
>
> Thank you very much for your time and effort in reviewing our paper and for your fruitful suggestions. We are very happy to read your positive assessment.
>
>
> **Weaknesses**
>
> >... the results apply primarily to “well-behaved” systems (smooth, band-limited, and not too chaotic). Real-world temporal processes might violate these conditions (e.g., non-smooth or highly non-contractive dynamics).
>
> Indeed, the approximation rates in Theorem 4.6 require a certain degree of regularity. Similar types of regularity assumptions are typically needed also for classical neural network approximation results. On the other hand, our universal approximation results in Theorem 4.8 only require continuity of the system. We fully agree that it will be an important direction of future research to obtain approximation bounds for systems with high degrees of roughness or non-contractive dynamics. We added a sentence to the conclusion emphasising this point.
>
>
> > Minor weakness: the paper does not include any experimental or numerical simulation results to complement the theory. All results are analytical. This is a weakness in the context of a machine learning conference. Still, what I said is merely a comment, rather than a criticism.
>
> Our work is indeed primarily theoretical, and its core contribution is to establish new analytical bounds and to clarify their behavior in comparison with existing theoretical guarantees in the literature (see our answer to the comment below in relation with classical counterparts). A variety of feedback architectures have already been analysed experimentally. In this context, the primary objective of our work is not empirical performance but rather the advancement of the theoretical understanding of the problem. We therefore focus on deriving bounds that can be directly compared, on equal footing, with prior theoretical results. We believe this comparison highlights the significance of our contribution and is the appropriate standard of evaluation for the type of result we provide. Extensive numerical evaluation experiments are part of our research agenda and will be executed in future work.
>
>
> >... Training a quantum model with many parameters is non-trivial, e.g., you might face barren plateaus, circuit noises, etc. You can argue that this is beyond the scope of this paper. Still, I think it is quite an important aspect to at least discuss them in the conclusion.
>
> We agree that it is important to comment on these points, thank you very much for pointing this out. We have added a discussion in the conclusion.
>
> **Questions**
>
> >... the potential for quantum advantage here, for instance, do there exist learning instances such that RQNNs and their classical counterparts exhibit a learning separation in terms of either time or sample complexity?
>
> Thanks a lot for this comment. We agree that it is important to compare to classical RNN counterparts. Indeed, the most similar setup with classical RNN has been considered in Theorem 3 of [Gonon, Grigoryeva, Ortega 2023]. While the approximation rates are the same in both results, the QRNNs offer the crucial advantage that the smoothness requirements are less stringent. In other words, the regularity requirements on the target systems are *strictly weaker* using QRNNs as opposed to classical RNNs. For example, for Sobolev function approximation classical RNNs require Sobolev regularity $N+d+3$ while QRNNs only requires $\frac{N+d}{2}+4$. While this is not a quantum advantage in the classical sense, this result still shows that QRNN provide an advantage in the sense that they are able to approximate a broader class of state-space systems. A paragraph emphasizing this kind of considerations has been added to the text, right under Theorem 4.6.
>
>
>
> > Have the authors considered the practical side of how one would train or set the parameters for an RQNN to approximate a given system? Would one use gradient-based variational quantum circuits training to fit observed data from the target system?
>
> Thank you for raising this point. Indeed, we would expect that the most effective way to train QRNN parameters would be gradient-based variational quantum circuit training to minimize the loss between the circuit predictions and the observed target data.  However, developing effective learning algorithms will need to be explored further in future work. Alternative approaches are presented in, for instance, Supplementary Section 1 of M. Cerezo et al., Nat Rev Phys 3, 625–644 (2021). We now comment on this in the conclusion.
>
>
> We hope that these clarifications and adjustments have helped to address all raised questions.

---

### Author Response · Authors · 2025-11-19
**General reply to reviewer comments**

Dear Reviewers

We would like to thank you for the time taken to carefully assess our paper and provide valuable feedback. We are very happy to hear your positive assessment of the paper.
Based on your comments, we have uploaded a revised version of the paper in which we have incorporated all your comments and concerns.
Below, we provide answers to each of your comments and questions individually. To make it easier to follow the changes, we marked in colour all changes with respect to the original submission. We hope that these replies clarify all your questions.
Please do not hestiate to let us know in case any of our answers require further clarification.

---

### Meta-Review · Area_Chair_DX8c · 2026-01-07

**Summary:**

The rebuttal addressed the concerns raised by the reviewers and provided a comprehensive analysis of the proposed method. One reviewer provided Strong Accept, two reviewers provided Accepts, and one with marginally above the acceptance threshold. The authors actively provided detailed answers.
The final version should include all reviewer comments, suggestions, and additional information from the rebuttal.

**Reviewer Concerns:**

All the reviewers' concerns have been addressed.

**Reviewer Scores:**

The reviewers' scores reflect the good quality of this submission.

---

### Decision · Program_Chairs · 2026-01-26

Accept (Poster)